# Multiple carbon cycle mechanisms associated with the glaciation of Marine Isotope Stage 4

James A. Menking [1] ✉, Sarah A. Shackleton[2], Thomas K. Bauska [3], Aron M. Buffen[1], Edward J. Brook[1], Stephen Barker [4], Jeffrey P. Severinghaus[2], Michael N. Dyonisius[5] & Vasilii V. Petrenko[6]

Here we use high-precision carbon isotope data ($\delta^{13}$C-$CO_2$) to show atmospheric $CO_2$ during Marine Isotope Stage 4 (MIS 4, ~70.5-59 ka) was controlled by a succession of millennial-scale processes. Enriched $\delta^{13}$C-$CO_2$ during peak glaciation suggests increased ocean carbon storage. Variations in $\delta^{13}$C-$CO_2$ in early MIS 4 suggest multiple processes were active during $CO_2$ drawdown, potentially including decreased land carbon and decreased Southern Ocean air-sea gas exchange superposed on increased ocean carbon storage. $CO_2$ remained low during MIS 4 while $\delta^{13}$C-$CO_2$ fluctuations suggest changes in Southern Ocean and North Atlantic air-sea gas exchange. A 7 ppm increase in $CO_2$ at the onset of Dansgaard-Oeschger event 19 (72.1 ka) and 27 ppm increase in $CO_2$ during late MIS 4 (Heinrich Stadial 6, ~63.5-60 ka) involved additions of isotopically light carbon to the atmosphere. The terrestrial biosphere and Southern Ocean air-sea gas exchange are possible sources, with the latter event also involving decreased ocean carbon storage.

Atmospheric $CO_2$ and Antarctic temperature were closely coupled over the last 800,000 years[1,2], and the importance of $CO_2$ in glacial cycles is widely recognized[3]. Many studies focus on the cause of the 80 ppm $CO_2$ rise during the last deglaciation[4–6], often highlighting the importance of a single mechanism or a single region. Processes that led to a glacial drawdown of $CO_2$ have received less attention, despite their importance for understanding glacial–interglacial $CO_2$ cycles. It is widely believed that $CO_2$ was sequestered in the deep ocean during glacial times via a combination of physical and biological mechanisms[7–9], but establishing the timing and importance of different processes has been challenging. About 40% of the total interglacial to glacial $CO_2$ decrease occurred during the Marine Isotope Stage 5-4 transition (MIS 5-4; ~72–67 ka), a period of global cooling and glacial inception. This transition was marked by the expansion of Southern Hemisphere glaciers and ice sheets[10], ocean cooling[8,11,12], decreasing sea level[13,14], and significant reorganization of ocean circulation[15–19].

The stable isotopic composition of $CO_2$ ($\delta^{13}$C-$CO_2$) in ancient air trapped in ice cores can trace processes in the carbon cycle that impact atmospheric $CO_2$ concentration[20,21]. Ice core $\delta^{13}$C-$CO_2$ data now exist for the entire period spanning the penultimate deglaciation (~150 ka) to the late Holocene (last 1 ka)[22–25] with high-resolution and high-precision data for some intervals (<200 yr, $1\sigma = 0.02$‰)[4,26–28]. Existing data spanning the MIS 5-4 transition are broadly consistent with $CO_2$ sequestration due to a more efficient marine biological pump[22], however, the resolution and precision of the existing data preclude more detailed interpretation.

Here we report high-resolution $\delta^{13}$C-$CO_2$ data spanning the MIS 5-4 transition, as well as Dansgaard–Oeschger (DO) event 19, and the gradual rise of $CO_2$ during the transition out of MIS 4 (Heinrich Stadial 6).

[1]College of Earth, Ocean, and Atmospheric Sciences, Oregon State University, Corvallis, OR 97331, USA. [2]Scripps Institution of Oceanography, University of California San Diego, La Jolla, CA 92093, USA. [3]British Antarctic Survey, Cambridge CB3 0ET, UK. [4]School of Earth and Environmental Sciences, Cardiff University, Cardiff CF10 3AT, UK. [5]Ice, Climate, and Geophysics, Niels Bohr Institute, 2200 Copenhagen, Denmark. [6]Department of Earth and Environmental Sciences, University of Rochester, Rochester, MI 14627, USA. ✉e-mail: James.Menking@utas.edu.au

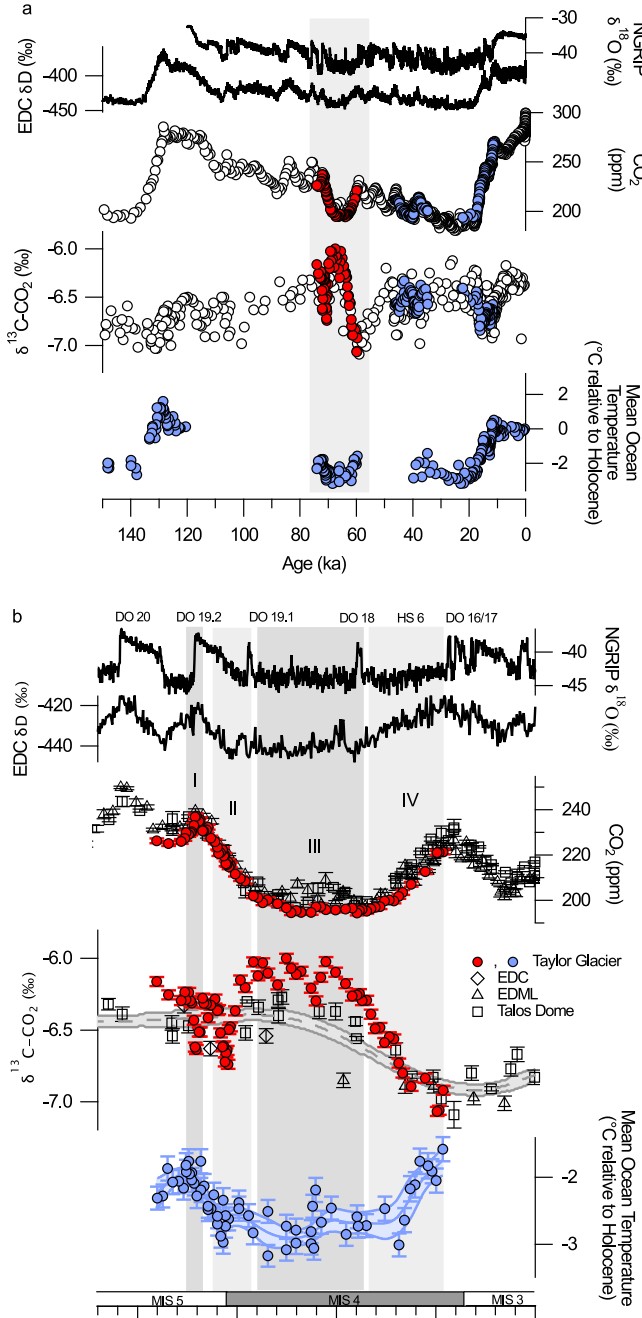

**Fig. 1 | High-resolution CO₂ and δ¹³C-CO₂ data from Taylor Glacier. a** Data from this study (red circles) show larger variations in δ¹³C-CO₂ across the transition into and out of Marine Isotope Stage 4 (74–59.5 ka) than preexisting δ¹³C-CO₂ data (white and blue circles) spanning the last two deglaciations (140–125 and 21–11 ka)[4,22–24] and the Heinrich Stadial (HS)-4/Dansgaard–Oeschger (DO)-8 transition (46–36 ka)[26]. The large changes in the δ¹³C-CO₂ are surprising given the relatively smaller magnitude changes in CO₂. Mean ocean temperature data derived from ice core noble gas measurements show relatively smaller changes across the MIS 5-4 transition[12] relative to the last two deglaciations[80–82]. North Greenland Ice Core Project (NGRIP)[48] and EPICA Dome C (EDC)[83] water isotopes are plotted for chronological and climatic reference. **b** An enlarged plot of the interval 77–55 ka compares Taylor Glacier data to preexisting CO₂ and δ¹³C-CO₂ data from EPICA Dome C (EDC), EPICA Dronning Maud Land (EDML), and Talos Dome ice cores. The interval is divided into four subintervals (I–IV) highlighting distinct modes of change in CO₂ and δ¹³C-CO₂ discussed in the text. Error bars represent 1-sigma analytical uncertainty.

We present a modeling framework for interpreting the results and discuss the likely causes of CO₂ evolution from 74.0 to 59.6 ka.

## Results and discussion

### Evolution of CO₂ and δ¹³C-CO₂ between 74.1–59.6 ka

The magnitude of isotopic changes in our data is larger than those observed during the last two deglaciations despite smaller changes in CO₂[4,23,24] (Fig. 1a). Taken at face value, the δ¹³C-CO₂ data suggest a complex evolution of the carbon cycle between 74.0–59.6 ka that was strongly influenced by processes with high leverage on δ¹³C-CO₂ relative to CO₂.

Broadly, we resolve multi-millennial changes in δ¹³C-CO₂ that are anti-correlated with CO₂ concentration (Fig. 1b). The most prominent example is the association of low CO₂ during MIS 4 with high δ¹³C-CO₂, including values >−6.00‰ (interval III in Fig. 1b), the most enriched observed over the last 140 ka. Similarly, when CO₂ concentration increased by 26 ppm at the end of MIS 4, δ¹³C-CO₂ decreased to −7.07‰, the most depleted value of the last 140 ka (interval IV in Fig. 1b). This feature marks the most enigmatic change in δ¹³C-CO₂ during the past 140 ka[22] with such depleted values not reached again until the mid-19th century due to the burning of ¹³C-depleted fossil fuels.

We also resolve fast changes in δ¹³C-CO₂ with varying relationships to CO₂ concentration. For example, there are two negative excursions in δ¹³C-CO₂ beginning at 72.5 and 71.1 ka (intervals I and II, respectively, in Fig. 1b). In the first isotopic excursion, CO₂ increased by 7 ppm while δ¹³C-CO₂ abruptly decreased by 0.4‰. The onset of this event is coincident with the Dansgaard–Oeschger (DO)-19 transition, an abrupt Northern Hemisphere warming at the end of MIS 5. The relative timing with respect to DO-19 is tightly constrained by the phasing of CH₄ variations measured in the same ice core, which closely tracked DO events[29]. In the second isotopic excursion, CO₂ decreased at a nearly continuous rate between 71 and 69 ka, but δ¹³C-CO₂ decreased by about 0.5‰ to a local minimum centered on 70.5 ka. This feature occurred early in the CO₂ drawdown associated with the MIS 5-4 transition and represents the only significant period where CO₂ and δ¹³C-CO₂ were positively correlated. δ¹³C-CO₂ subsequently recovered to pre-excursion values, then continued to increase to −6.0‰ by 69 ka. The full increase in δ¹³C-CO₂ from 70.5 to 69 ka was 0.71‰, while the net enrichment above the pre-excursion value at 71.1 ka was about 0.26‰. We also observe variations up to 0.3‰ in δ¹³C-CO₂ during MIS 4 without significant changes in CO₂.

### Using δ¹³C-CO₂ to understand CO₂ changes between 74.1–59.6 ka

As a heuristic tool, we compiled perturbations to carbon cycle models and examined the results on a cross-plot of δ¹³C-CO₂ and CO₂ concentration[4,20,22,30] (Fig. 2a). We compiled results from the OSU 14-box model[4] (Supplementary Fig. 1), as well as the Box Model of the Isotopic Carbon Cycle (BICYCLE)[20,22,30], the Bern 3d Earth System Model[31,32], the University of Victoria Earth System Climate Model (UVic ESCM)[33,34], and the LOVECLIM isotope-enabled earth system model of intermediate complexity[35] (Supplementary Fig. 2). We grouped the results from all models into six broad categories— sea surface temperature, biological pump (i.e., productivity and circulation), sea ice, land carbon, alkalinity, and Southern Ocean gas exchange, representing the primary drivers of glacial–interglacial and millennial-scale CO₂ change discussed in the literature[4,36,37]. A drop in CO₂ coincident with a decrease in δ¹³C-CO₂ indicates uptake from a cooling ocean, whereas a drop in CO₂ and increase in δ¹³C-CO₂ indicates uptake by a depleted carbon reservoir (e.g., organic material sequestered on land or exported to the deep ocean). Simple modeling suggests that a drop in CO₂ coincident with a very large increase in δ¹³C-CO₂ would indicate a major reduction in Southern Ocean air–sea gas exchange driven by increased Antarctic sea ice extent or decreased wind stress[4]. Box

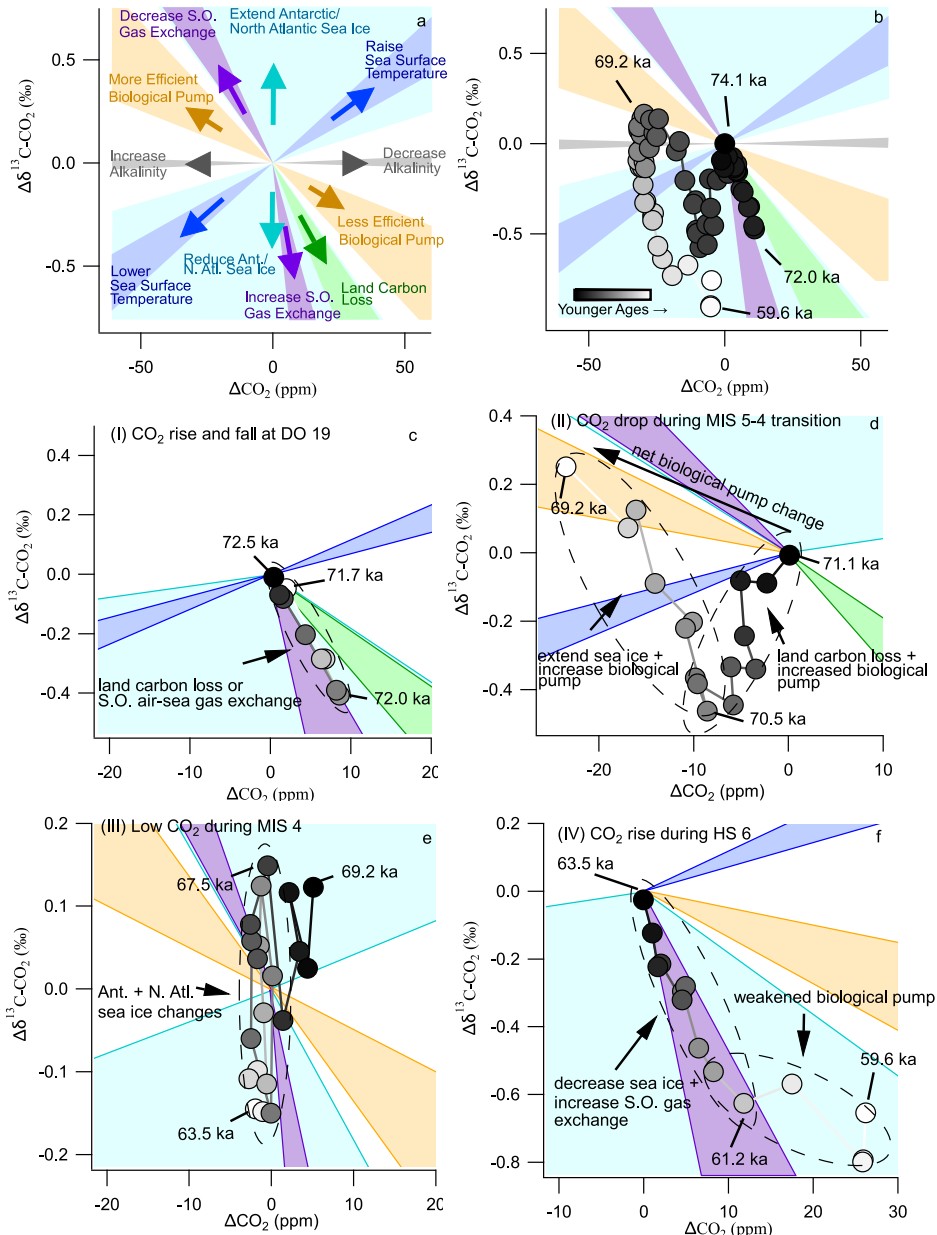

**Fig. 2 | Model framework for interpreting CO₂ and δ¹³C-CO₂ data. a** Compilation of model results estimating the change in δ¹³C-CO₂ per unit change in CO₂ concentration due to different processes, indicated by shaded regions (see Supplementary information for details). The shading for the processes relevant to our interpretations in each interval is drawn on the following panels. **b** The temporal evolution of δ¹³C-CO₂ and CO₂ data is indicated by the color gradient on the markers. **c–f** δ¹³C-CO₂ and CO₂ data from each of the four intervals I–IV shown in Fig. 1. Note the axes are scaled differently for each panel. **c** δ¹³C-CO₂ and CO₂ change during Dansgaard-Oeschger (DO) 19 (interval I in Fig. 1). The data are most consistent with an increase in Southern Ocean air–sea gas exchange rates or a release of land carbon. **d** δ¹³C-CO₂ and CO₂ data for the negative isotope excursion and

enrichment during the Marine Isotope Stage (MIS) 5-4 transition (interval II in Fig. 1). The negative excursion is consistent with a large pulse of land carbon combined with increasing efficiency of the biological pump. The growth of Antarctic sea ice and continued deep carbon storage could explain the following enrichment trend. **e** Oscillations in δ¹³C-CO₂ during MIS 4 were accompanied by very little change in CO₂ concentration (interval III in Fig. 1), perhaps due to fluctuations in Antarctic sea ice. **f** The δ¹³C-CO₂ and CO₂ change during Heinrich Stadial (HS) 6 (interval IV in Fig. 1). The large decrease in δ¹³C-CO₂ is consistent with decreasing Antarctic sea ice and increased air–sea gas exchange in the Southern Ocean. The youngest data (60.9–59.6 ka) are consistent with decreasing efficiency of the biological pump.

models have shown that combined increases in the extent of northern and southern sea ice could produce a large increase in δ¹³C-CO₂ with a canceling effect on CO₂ concentration, but this result is yet to be reproduced in more complex models[20] (Supplementary information). Alternatively, a drop in CO₂ with little change in δ¹³C-CO₂ could indicate a CO₂ sink dominated by changes in the CaCO₃ cycle (e.g, the weathering of CaCO₃ on land or dissolution of CaCO₃ in marine sediments).

Although different models predict slightly different relationships between CO₂ and δ¹³C-CO₂, changes due to individual processes are generally distinct and consistent among models (Fig. 2a). More complex perturbations involving multiple processes may be estimated as linear combinations of single perturbations[22]. The system is under-constrained as we have two knowns (CO₂ and δ¹³C-CO₂), and (broadly) four unknown sources/sinks in the form of changes in ocean temperature, organic carbon storage, CaCO₃, and air–sea gas exchange.

However, we can effectively remove two degrees of freedom in the system by (1) employing coeval constraints on mean ocean temperature provided by noble gas measurements[12], and (2) ruling out any changes due to the slow response of the $CaCO_3$ cycle (e.g. weathering, reef building, dissolution/burial) for rapid variations in $CO_2$. We divided the data into four intervals based on the features in the $CO_2$ concentration data. Below we discuss possible explanations for the observed changes during each interval considering the patterns in atmospheric $CO_2$ and $\delta^{13}C$-$CO_2$ predicted by the carbon cycle models as well as additional constraints on the timing of oceanic processes from paleoceanographic data. This approach requires that the interpretation is largely limited to qualitative descriptions of whether or not a process, or combinations of processes, are driving changes in $CO_2$. In the Supplementary information, we present forward model simulations with the OSU box model that demonstrate how the sequence of changes in atmospheric $\delta^{13}C$-$CO_2$ and $CO_2$ might have been enacted by the carbon cycle changes proposed in the main text and highlight intervals where the data are difficult to reproduce.

## $CO_2$ increase and isotopic excursion at DO-19

Previous work highlighted a contrast between millennial-scale $CO_2$ changes during MIS 5 versus MIS 3, with local maxima in $CO_2$ occurring closer to the onset of DO events during MIS 5[38]. High-resolution $CO_2$ data spanning MIS 3 and the last deglaciation (60–11 ka) now show many instances of abrupt $CO_2$ increases that are in phase with Greenland warming[39,40], though the magnitude of the $CO_2$ increases appears to be smaller than at DO-19 (72.1 ka). Furthermore, the limited high-resolution $\delta^{13}C$-$CO_2$ data accompanying MIS 3 DO events do not show negative excursions[4,26] (Fig. 3). The feature at DO−19 resolved by the data may imply that the millennial response of the carbon cycle to Northern Hemisphere warming was different during MIS 5a versus MIS 3 or the last deglaciation. The $CO_2$ increase would require processes that are in phase with Northern Hemisphere warming and that have a large, negative effect on $\delta^{13}C$-$CO_2$ per unit increase in $CO_2$ (Fig. 2c).

The cross-plot suggests the changes in $\delta^{13}C$-$CO_2$ and $CO_2$ across DO-19 (72.1 ka) are consistent with increased Southern Ocean gas exchange (Fig. 2c), which could have resulted from a shift in the strength or position of the Southern Hemisphere Westerlies[41]. Low sea ice coverage (Fig. 4e) would have supported increased gas exchange.

South Atlantic opal data show an upwelling event occurred near the onset of DO-19 (Fig. 4g)[42], which may indicate shifting Westerlies, but we note that the age model for the opal data likely cannot resolve whether this was really in-phase with Greenland warming. Further, the original mechanism that this interpretation is based on would have the upwelling occurring when Antarctica was warming during the stadial preceding DO-19[42]. Another plausible explanation for the DO-19 $CO_2$ excursion is that Northern Hemisphere warming caused a transfer of terrestrial carbon to the atmosphere and ocean. This seems at odds with studies suggesting the terrestrial response to warming is regrowth (lowering atmospheric $CO_2$)[36,43,44], but we note one model shows a rapid increase in $CO_2$ immediately after AMOC recovery because increased soil-respiration rates at mid-latitudes temporarily exceed slower regrowth of boreal forests[45]. The rapid $CO_2$ increase and large $\delta^{13}C$-$CO_2$ decrease at DO-19 may be an example of this mode of change, but the stark contrast in $\delta^{13}C$-$CO_2$ across DO-19 versus DO-8 (Fig. 3) would imply that the response of the terrestrial biosphere to AMOC recovery was different at different times. One reason might be that the terrestrial biosphere was larger during MIS 5a relative to MIS 3, and therefore sudden changes in Northern Hemisphere temperature had a greater impact on terrestrial carbon storage. The MIS 5a data, better suited as analogues for today than data from the last glacial period, may therefore suggest that positive climate-carbon feedbacks operating in the Northern Hemisphere are larger than previously indicated[26]. Alternatively, the events during MIS 3 may have been convolved with larger sea surface temperature changes that masked the $\delta^{13}C$-$CO_2$ signature of land carbon release[39]. A third plausible mechanism for the DO-19 data is that the reinvigoration of the AMOC flushed out stagnant deep Atlantic water that was rich in respired carbon[38]. Such a transfer of respired carbon from the deep ocean would normally be associated with circulation and/or productivity changes that alter the efficiency of the ocean biological pump, but the cross-plot suggests a mechanism that plots more steeply than ocean biological pump changes (Fig. 2c). Therefore, the implication of the isotope excursion would be that either (1) there was more respired carbon accumulated in the deep ocean during Greenland Stadial 20 (GS-20), the cold period that occurred ~73 ka immediately prior to the onset of DO-19, than during MIS 3 stadials, or (2) AMOC switch-on was stronger in late MIS 5 than MIS 3. The former is plausible considering

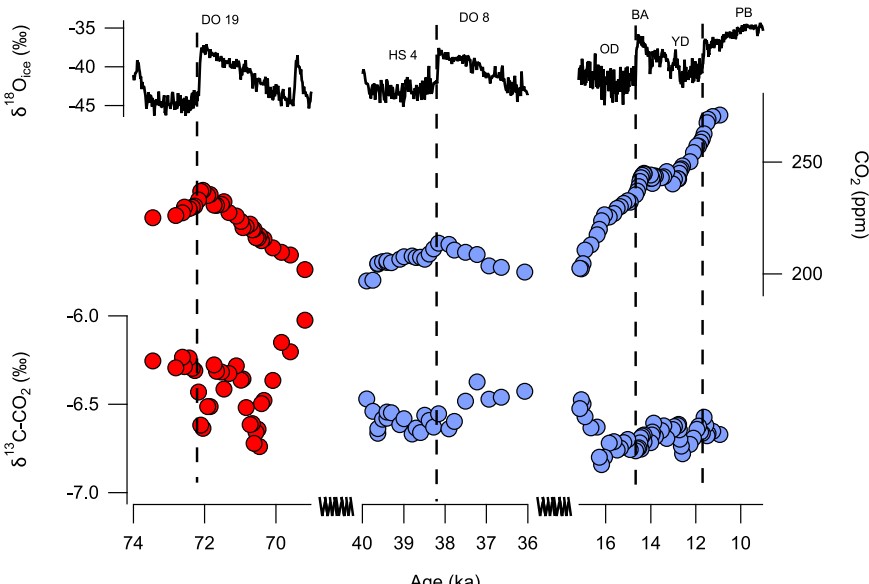

**Fig. 3 | Comparison of abrupt $CO_2$ changes.** The negative isotopic excursion associated with the $CO_2$ increase at Dansgaard–Oeschger (DO)-19 (red circles, this study) did not occur at other Northern Hemisphere warming events with similar fast $CO_2$ increases, e.g. DO-8 or the Oldest Dryas (OD)–Bølling–Allerød (BA) transition (blue circles)[4,26]. Water stable isotope data are from North Greenland Ice Core Project (NGRIP)[48].

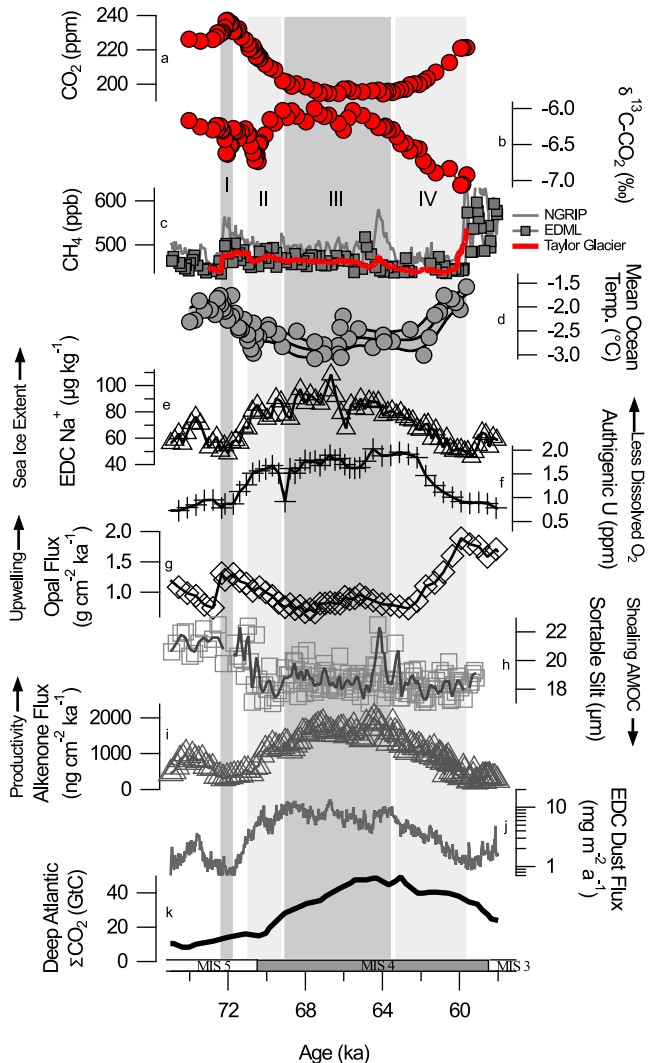

**Fig. 4 | Comparison of Taylor Glacier data to other paleoclimate reconstructions.** From top to bottom we show **a** Taylor Glacier $CO_2$, **b** Taylor Glacier $\delta^{13}C$-$CO_2$, **c** Taylor Glacier $CH_4$ with comparison to EPICA Dronning Maud Land (EDML) and North Greenland Ice Core Project (NGRIP) $CH_4$[84,85], **d** mean ocean temperature derived from noble gas measurements of Taylor Glacier ice cores[12], **e** EPICA Dome C (EDC) Na+ proxy for Antarctic sea ice extent[57], **f** South Atlantic TN057–14PC authigenic U proxy for deep ocean oxygenation[50], **g** South Atlantic TN057–14PC opal flux[42], **h** Western Boundary Undercurrent sortable silt circulation tracer[16], **i** South Atlantic TN057-21 alkenone flux productivity proxy[52], **j** EDC dust flux[59], and **k** Deep Atlantic total dissolved inorganic carbon reconstruction[49]. Age models are consistent with AICC 2012[11,29,86].

that GS-20 is thought to have been extremely cold due to the Toba eruption[46–48]. The impact of the cooling on the terrestrial carbon cycle may have allowed more isotopically light carbon to accumulate in the deep ocean. To summarize, our observations at DO-19 suggest a new type of centennial-scale variability at the onset of interstadials that requires an isotopically depleted source of $CO_2$, but it is not well understood if the $CO_2$ is ultimately sourced from the ocean or the land.

## $CO_2$ decrease at the onset of MIS 4
We interpret the enriched $\delta^{13}C$-$CO_2$ values during MIS 4 to represent increased storage of carbon in the deep ocean. This view is supported by South Atlantic proxy records including the B/Ca proxy for total dissolved inorganic carbon[49] (Fig. 4k), the authigenic U proxy for deep water oxygenation[50] (Fig. 4f), and South Atlantic benthic $\delta^{13}C$[51](not shown), pointing to higher storage of respired carbon during MIS 4

relative to MIS 5a. Proxies for productivity and iron fertilization show that organic carbon export increased in the Subantarctic South Atlantic (Fig. 4i–j)[52–54], while ocean circulation proxies show a shoaling of Atlantic Meridional Ocean Circulation (AMOC) and expansion of Antarctic Bottom Water (AABW)[16,19,49,55] (Fig. 4h). The development of this two-celled structure of glacial water masses is believed to enhance deep ocean carbon storage[15,56].

However, the negative $\delta^{13}C$-$CO_2$ anomaly centered at 70.5 ka (Fig. 1b, interval II) demonstrates that the evolution of the carbon cycle across the MIS 5-4 transition was more complex than simply increasing the efficiency of the biological pump. The $\delta^{13}C$-$CO_2$ anomaly is a difficult feature to explain and is not reflected in other paleoceanographic data (Fig. 4). Cooling sea surface temperatures were partly responsible for the negative isotope trend because increasing $CO_2$ solubility causes decreases in both $CO_2$ and $\delta^{13}C$-$CO_2$ (Fig. 2a). Although a cooling ocean contributed to the MIS 5-4 $CO_2$ drawdown across the entire interval from 72.5 to 67 ka (9 ± 3 ppm decrease[12] with attendant $\delta^{13}C$-$CO_2$ decrease of ~0.1‰), this does not explain the isotope anomaly between 71 and 70.5 ka because the $\delta^{13}C$-$CO_2$ change is too large (Fig. 2d). Furthermore, mean ocean temperature data derived from noble gas measurements on the same ice core indicate that most of the ocean cooling probably occurred before the negative isotope excursion[12] (Fig. 4d), unless there was a very rapid and large cooling anomaly centered at 70.5 ka that is not resolved given the resolution and uncertainty of the mean ocean temperature data. We also rule out $CaCO_3$ compensation as a significant player in the $CO_2$ drawdown because (1) $CaCO_3$ compensation operates on a multi-millennial timescale, and (2) the predicted pattern of $CO_2$ and $\delta^{13}C$-$CO_2$ changes (change in $CO_2$ accompanied by little to no change in $\delta^{13}C$-$CO_2$) is not evident in the data (Fig. 2d). If $CO_2$ decrease was due to slow $CaCO_3$ compensation in response to prior events that occurred during MIS 5a, it would have been masked by larger carbon cycle changes during the MIS 5-4 transition and was not likely to contribute more than 4 ppm to the $CO_2$ decline (supplementary information). Additional mechanism(s) during the $CO_2$ drop is therefore needed to explain the observed $\delta^{13}C$-$CO_2$ depletion. One possibility is that a pulse of isotopically light $CO_2$ to the atmosphere, perhaps from land, was roughly balanced by increasing $CO_2$ uptake via enhanced Subantarctic Ocean biological productivity or Antarctic sea ice extension between 71.1 and 70.5 ka, creating a net decline in $CO_2$ and decreasing $\delta^{13}C$-$CO_2$ (Supplementary information). In this scenario, the addition of light carbon was complete by 70.5 ka when $\delta^{13}C$-$CO_2$ began to recover to pre-excursion values. Increases in the efficiency of the biological pump and Antarctic sea ice coverage could have sequestered $CO_2$ in the deep ocean, which lowered $CO_2$ and raised $\delta^{13}C$-$CO_2$ beyond the values prior to the excursion. Proxy data support this scenario because they show an extension of sea ice and shifts in ocean circulation and marine productivity that enhanced carbon sequestration starting ~70.5 ka[16,49,50,52,57] (Fig. 4e–k). We suggest cooling and drying of boreal forests during the descent into MIS 4 could have been the source of the land carbon, possibly combined with remobilization of carbon on continental shelves as the sea level dropped. This hypothesis is attractive because it combines processes that were probably active during the transition into MIS 4, however, the magnitudes of perturbations required in simple forward model simulations to reproduce the data are quite large, and we are unable to precisely quantify the changes in land carbon (Supplementary information).

## Low $CO_2$ and $\delta^{13}C$-$CO_2$ variations during MIS 4
Although $CO_2$ remained low and stable during MIS 4, $\delta^{13}C$-$CO_2$ varied between −5.99‰ and −6.30‰ (Fig. 1b, interval III). Changes in $\delta^{13}C$-$CO_2$ of this magnitude with little to no change in $CO_2$ mark a unique mode of variability not previously documented in ice core $\delta^{13}C$-$CO_2$ records[4]. The implication is that processes in the carbon cycle can be active and yet cause zero net change in atmospheric $CO_2$ concentration. Simple

models show that very large changes in $\delta^{13}C\text{-}CO_2$ may result from changes in Southern Ocean gas exchange rates[4,20], which could arise due to shifts in the strength or position of the Westerlies and/or changes in the extent of Antarctic sea ice coverage, the latter with a potential canceling effect on $CO_2$ if combined with changes in North Atlantic sea ice coverage[20,58]. Opal flux data do not implicate significant changes in the Westerlies during MIS 4, but ice-core $Na^+$ data suggest variations that can be linked to sea ice extent between 69 and 64 ka (Fig. 4e). There are also changes in Southern Ocean dust flux observed in ice cores[59] (Fig. 4j) and sediment cores[53] (not shown) during MIS 4, which may have modulated Subantarctic biological productivity, but the cross-plots suggest there must have been some compensating mechanism to offset changes in $CO_2$ if the biological pump is invoked (Fig. 2e). It is tempting, therefore, to invoke northern and southern sea ice changes, but we caution that the box model results showing large $\delta^{13}C\text{-}CO_2$ changes are likely highly dependent on model architecture and the degree to which the zones of deepwater formation are out of equilibrium with the atmosphere[20], and to our knowledge have not been reproduced in more complex models[60]. We also note the timing of fluctuations in $Na^+$ (or dust) is not precisely aligned with the features observed in $\delta^{13}C\text{-}CO_2$.

Lastly, we note that $\delta^{13}C\text{-}CO_2$ decreased by 0.24‰ beginning at 67 ka, about three thousand years before the start of the $CO_2$ rise associated with Heinrich Stadial 6 (Fig. 1b). We consider this initial depletion the same mode of variability as the $\delta^{13}C\text{-}CO_2$ fluctuations during MIS 4 accompanied by virtually no $CO_2$ change.

## $CO_2$ rise and $\delta^{13}C\text{-}CO_2$ decrease during the MIS 4-3 transition

The large $\delta^{13}C\text{-}CO_2$ depletion and $CO_2$ increase at the end of MIS 4 (shaded interval IV in Fig. 1b) represents an unprecedented and enigmatic mode of variability relative to the last 150 ka (Fig. 1a). Between 63.5 and 60 ka, $CO_2$ slowly rose and $\delta^{13}C\text{-}CO_2$ decreased by 0.8‰, implicating a release of isotopically light carbon to the atmosphere. It is worth emphasizing that the feature represents the largest magnitude decrease in $\delta^{13}C\text{-}CO_2$ in the ice core record, exceeded only in magnitude by the decrease in $\delta^{13}C\text{-}CO_2$ observed during the industrial era due to the combustion of fossil fuels. It is also notable that the changes observed in other paleoceanographic records between 63.6 and 60 ka mostly do not reflect the exceptional nature of the $\delta^{13}C\text{-}CO_2$ change (Fig. 4), which makes the $\delta^{13}C\text{-}CO_2$ feature difficult to explain. Our interpretations during this interval are similar to Eggleston et al.[22] in that the $CO_2$ rise during Heinrich Stadial (HS)-6 was dominated by a relaxation of the biological pump and ventilation of deep water via Southern Ocean upwelling, but our more highly resolved data add more nuance regarding the timing. Proxy data suggest a decrease in the efficiency of the biological pump between 64 and 58 ka[16,49,50,52] (Fig. 4f–k), but the cross-plot suggests that the change in $\delta^{13}C\text{-}CO_2$ was too large to be due solely to biological pump changes, except perhaps for the interval between 61.2 and 59.6 ka (Fig. 2f). Enhanced Southern Ocean air–sea gas exchange is consistent with the steeper trend between 63.5 and 61.2 ka. We suggest the following sequence of events occurred: (1) enhanced storage of respired carbon during MIS 4 primed the deep ocean with isotopically light carbon prior to 63.5 ka, and (2) the strength of the southern hemisphere westerlies increased and/or they shifted south as Antarctica warmed during Heinrich Stadial (HS) 6[61]. This mechanism is supported by South Atlantic opal data showing a large increase in opal flux near the end of HS-6[42] (Fig. 4g). Depending on the age model used, the increase in opal flux lags the change in $CO_2$ and $\delta^{13}C\text{-}CO_2$ by up to 3 millennia, but this is likely consistent with our interpretation as model simulations of shifts in the Westerlies predict such a delay in opal accumulation relative to the winds[31]. Decreasing the Antarctic sea ice extent (Fig. 4e) could have also enhanced Southern Ocean air–sea gas exchange. (3) Lastly, the continued waning of deep ocean carbon storage due to relaxation of the ocean's biological pump or increased deep ocean ventilation

between 61.2 and 60.0 ka can explain an additional 14 ppm $CO_2$ rise and 0.17‰ decrease in $\delta^{13}C\text{-}CO_2$. The mechanisms invoked to explain the $CO_2$ rise across the MIS 4-3 transition is not unlike those that explain the rise in $CO_2$ across the last deglacial transition[4,24]. One key difference between the two intervals, and a plausible explanation for why the MIS 4-3 change in $\delta^{13}C\text{-}CO_2$ was so great, is that the carbon cycle changes were less convolved with the impact of rising sea surface temperature compared to the deglaciation. Ocean heating is estimated to have contributed only -10 ppm to the $CO_2$ rise at the MIS 4-3 transition[12], but contributed -30 ppm during the last deglaciation, which would partially cancel the impact on $\delta^{13}C\text{-}CO_2$ of a relaxed biological pump or enhanced Southern Ocean gas exchange.

## Concluding remarks

$\delta^{13}C\text{-}CO_2$ and $CO_2$ data constrain carbon cycle variability across the MIS 5-4 transition, during MIS 4, and the transition into MIS 3. A single process was not solely, or even dominantly, responsible for controlling atmospheric $CO_2$. Rather, the data show that the climate changes associated with the descent into and out of MIS 4 triggered a succession of different carbon cycle processes that conspired to alter $CO_2$. The data are consistent with a more efficient biological pump and increased carbon storage in the MIS 4 deep ocean, but large and fast variations in $\delta^{13}C\text{-}CO_2$ that were previously not observed in ice core data implicate the superposition of rapid land carbon transfers and/or shifts in Southern Ocean air–sea gas exchange rates (perhaps modulated by sea ice) on the drawdown of $CO_2$ into the ocean. The data also demonstrate that processes were active during MIS 4 that altered $\delta^{13}C\text{-}CO_2$ with little to no change in $CO_2$ concentration. The data document a mode of rapid $CO_2$ variability associated with Northern Hemisphere warming at the onset of DO-19 characterized by net additions of light carbon to the atmosphere, which is distinct from similar events observed during the later part of the last glacial period. The result may suggest that a previous study that concluded positive climate-carbon feedbacks were small during abrupt warmings needs further examination using better-suited climatic analogs[26]. Forward simulations highlight the exceptional nature of the variations resolved in the data and demonstrate that, while difficult to reproduce exactly, the majority of the variability in $CO_2$ and $\delta^{13}C\text{-}CO_2$ can be explained with the right sequence of mechanisms. Future modelling work should explore the hypotheses proposed in this manuscript using the $\delta^{13}C\text{-}CO_2$ data as a constraint.

## Methods

### Field site and sample collection

Samples for this study were retrieved from the Taylor Glacier ablation zone. Taylor Glacier is an outlet glacier of the East Antarctic Ice Sheet that terminates in the McMurdo Dry Valleys. Relatively slow flow (-10 m yr$^{-1}$) and high ablation rates (up to -20 cm yr$^{-1}$) result in an -80 km ablation zone where old ice ranging in age from -130 to 7 ka outcrops in various locations[62–64]. In the 2014–2015 and 2015–2016 field seasons, ice cores were retrieved that contain the full MIS 5-4 transition in the ice and gas phases, as well as MIS 4 and much of the MIS 4-3 transition. The ice cores were retrieved with the Blue Ice Drill (BID)[65], a 24 cm diameter drill designed for retrieving large volume samples suitable for isotope analyses.

### Age model

The ice and gas bubbles were dated by matching variations in dust and $CH_4$, respectively, to preexisting ice core records tied to the Antarctic Ice Core Chronology (AICC) 2012[29,66]. The gas chronology used in this study was revised by matching variations in $CH_4$ concentration to similar variations in the NGRIP ice core (also tied to AICC 2012) and adopting two new tie points to match the $CO_2$ rise in the later part of the record. Relative age uncertainty with respect to the matching is 0.9 ka[29]. Age uncertainties are up to 2.5 ka if the absolute age

uncertainty of the AICC 2012 is considered. We note that age uncertainties do not greatly affect our interpretations because the cross-plot analyses are age-independent. The age uncertainty for the purpose of comparing $CO_2$ and $\delta^{13}C$-$CO_2$ is zero given that those measurements are made on the same air samples. The age uncertainty between $\delta^{13}C$-$CO_2$ and mean ocean temperature is also nearly zero since both were measured on samples from the same ice cores.

## Analytical and calibration procedures

Improved precision was achieved by using ancient air from large (250–500 g) ice core samples from the ablation zone of Taylor Glacier, Antarctica[4]. The data were produced using dual-inlet isotope ratio mass spectrometry and extraction and purification procedures developed at OSU[67]. The dataset represents a substantial improvement on existing data due to (1) higher time resolution (average resolution = 230 yr between 74.0 and 59.6 ka), (2) higher precision ($1\sigma$ = 0.03‰ on depth-adjacent replicate samples), and (3) sample collection without drill fluid, which is known to cause artifacts in isotope measurements despite cleaning protocols. The $\delta^{13}C$-$CO_2$ values are reported relative to VPDB.

Samples were cut vertically every 15 cm from 1/4 BID cores that were sampled in the field and stored at below −20 °C. The 15 cm sections were cut longitudinally into hexagonal prisms that typically measured 15 cm × 6 cm × 6 cm. The size of each hexagonal sample varied somewhat depending on core quality. The outer surfaces were cleaned further with a ceramic blade. Sample mass ranged from 200 to 400 g and averaged 290 g. Samples with visible fractures were not used. A total of 84 discrete samples were measured for $\delta^{13}C$-$CO_2$ and $CO_2$ concentration at 67 discrete depths with 17 samples measured in replicate. The average depth and age spacing were 1 sample every 25 cm, or 1 sample every ~230 years on the gas age scale[29].

$\delta^{13}C$-$CO_2$ was measured by dual inlet isotope ratio mass spectrometry at Oregon State University[67]. Air was extracted from ice using a dry extraction method in which vacuum canisters with abrasive grating surfaces were shaken at −65 °C for 1 h. The estimated grating efficiency was 70–90% based on measuring the mass of the intact sample and the mass of the ungrated pieces of ice left in the canisters after shaking. A typical extraction yielded 22 cm³ STP of air. The $CO_2$ was purified using a −196 °C cryotrap cooled with liquid nitrogen. The trap consisted of a ¼″ outer diameter, stainless steel cold finger fitted to a Swagelok valve that allowed the apparatus to be sealed and disconnected manually from the vacuum line. The cold finger was subsequently attached to a dual inlet MAT 253 mass spectrometer fitted with a microvolume inlet, and $CO_2$ was transferred to the micro-volume at −196 °C. The $^{13}C/^{12}C$ ratio was measured against a pure $CO_2$ working reference gas (Oztech). The $\delta^{13}C$-$CO_2$ of the working reference was determined to be −10.51‰ relative to NBS-19. For each sample, a small aliquot of whole air was captured in a stainless-steel tube in a cryostat at 12 K prior to $CO_2$ separation. The air aliquot was analyzed for $CO_2$ concentration using an Agilent gas chromatograph with a Ni catalyst coupled to a flame ionization detector, similar to the system described by Ahn et al.[68]. $CO_2$ concentration measurements were calibrated to the WMO 2007 scale[69,70] by measuring standard air from Niwot Ridge, Colorado with known $CO_2$ concentration. The INSTAAR Stable Isotope Laboratory, Colorado calibrated the $\delta^{13}C$-$CO_2$ of the standard air to the VPDB-$CO_2$ scale by measuring it against NBS-19. Measuring the standard air against the Oztech working reference gas permitted one-point calibrations of ice core sample air measurements to the VPDB-$CO_2$ scale[67]. Care was taken to match the size of samples and standards to avoid introducing linearity artifacts.

Several corrections were applied to the measurements. $\delta^{13}C$-$CO_2$ was corrected for the isobaric interference of $N_2O$ by determining the $N_2O/CO_2$ ratio in samples[71]. This was accomplished by peak jumping to

monitor NO fragments at m/z 30 as sample air depleted from the microvolume at the end of the $\delta^{13}C$-$CO_2$ measurement[67]. The magnitude of this correction was 0.1–0.3‰ depending on the $N_2O$ concentration. The $N_2O/CO_2$ ratio allowed calculation of the $N_2O$ concentration once $CO_2$ was determined independently by gas chromatography. The correction for the isobaric interference of $^{17}O$ followed the formulation of Santrock et al.[72]. $\delta^{13}C$-$CO_2$ was corrected for gravitational fractionation in the firn column by subtracting the enrichment of $\delta^{15}N$-$N_2$[73] measured at Scripps Institution of Oceanography[74,75]. $\delta^{15}N$-$N_2$ was not measured for each $\delta^{13}C$-$CO_2$ depth interval, so the $\delta^{15}N$-$N_2$ data were interpolated linearly onto the $\delta^{13}C$-$CO_2$ depth scale to derive the gravitational correction at all depths. $CO_2$ concentration was also corrected for gravitational enrichment in the firn[76]. $CO_2$ and $\delta^{13}C$-$CO_2$ were corrected for a constant instrumental blank by measuring standard air introduced over gas-free ice (−1.5 ppm for $CO_2$ concentration and +0.066‰ for $\delta^{13}C$-$CO_2$)[67].

## Data quality

The precision for $\delta^{13}C$-$CO_2$, $CO_2$, and $N_2O$ measurements was estimated as the pooled standard deviation of replicate pair measurements (after rejecting four samples described below). The precision (1-sigma standard deviation of pooled replicate pairs) was 0.032‰ for $\delta^{13}C$-$CO_2$, 1.10 ppm for $CO_2$, and 3.60 ppb for $N_2O$. Replicate measurements are reported as averages.

Three results were rejected when leaks in the vacuum line or vacuum chambers occurred. These outliers were easily identifiable as large ($>2\sigma$) depletions in $\delta^{13}C$-$CO_2$ measured simultaneously with enrichments in $CO_2$ relative to adjacent samples, consistent with modern laboratory air mixing with the air extracted from the ice core samples. One additional sample was rejected because of anomalously high $N_2O$ concentration (30 ppb enriched relative to adjacent samples), which caused a bias in the $N_2O$ isobaric correction that resulted in poor $\delta^{13}C$-$CO_2$ replication. The reason for anomalously high $N_2O$ in this sample is unknown, though the dust concentration in this depth interval is relatively high, and in-situ production of an $N_2O$ artifact is possible in dusty ice[77–79]. Another possibility is a leak of $N_2$ (from lab air) into the mass spectrometer during sample handling that produces a NO fragment artefact.

## Data availability

The data generated in this study have been deposited in the United States Antarctic Program Data Center at https://www.usap-dc.org/view/dataset/601600.

## Code availability

At the time of publishing, the code for the OSU box model is being revised for a future publication that focuses on the model. The current version of the code is available by request from the corresponding author.

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

## Acknowledgements

We thank Kathy Schroeder and Mike Jayred for enormous assistance in the field. We also thank Mike Kalk for assistance at the OSU ice core lab. This project was funded by the National Science Foundation: US NSF PLR-1245821 EJB, US NSF PLR-1245659 VVP, US NSF PLR-1246148 JPS.

## Author contributions

J.A.M. made measurements on ice samples with assistance from A.M.B. J.A.M. performed all analyses and carbon cycle modeling with assistance from T.K.B. J.A.M. interpreted the data and analyses with input

from S.A.S., T.K.B., S.B., E.J.B., and J.P.S. J.A.M., S.A.S., T.K.B., M.N.D., J.P.S., and V.V.P. collaborated in fieldwork and ice core retrieval. E.J.B., V.V.P., and J.P.S. designed the study with input from T.K.B. J.A.M. wrote the manuscript with contributions from S.A.S., T.K.B., A.M.B., E.J.B., S.B., J.P.S., M.N.D., and V.V.P.

## Competing interests

The authors declare no competing interests.
