## [Peer Review File · Nature Communications]

REVIEWER COMMENTS

Reviewer #1 (Remarks to the Author):

This manuscript by Menking et al. presents new high resolution data on atmospheric CO₂ and δ¹³C_{CO₂ concentrations across the last interglacial-glacial transition. The combination of these two proxies, together with the high resolution and precision of the data allow the authors to examine in detail the combination of carbon cycle mechanisms that were at play during this climate transition. The manuscript employs a clever model-data comparison approach in order to attribute mechanisms or combinations of mechanisms to different time periods within the overall climate transition. I also appreciate how it links these 'cross plot' interpretations against multi-proxy datasets (fig 3).}

While the manuscript raises a lot of questions that it cannot answer, it is an important contribution as it shows how multiple mechanisms can act together to produce transitions in CO₂, and how these mechanisms can be different depending on the 'starting' conditions. The possibility that carbon feedbacks may be stronger than we thought, based on MIS 3 analogs, is an important message, and shows the carbon-climate relationship still has surprises in store. The dataset and hypotheses put forward will stimulate future research in this area.

The manuscript is well written and the figures are informative. I don't have any major concerns, just a number of suggestions and queries to improve the manuscript.

I think the biggest weakness of this study is the reliance on a compilation of model runs, which are not exactly the same in terms of their set up or perturbations. While arguably this makes the results more robust than using a single model, it does complicate the interpretation, and an argument could be made that it would be better to use a single model with a range of sensitivities.

The biological pump and SST perturbations are fairly straightforward, but there is a lot of overlap in the land carbon/sea ice/gas exchange mechanisms, and the text is confusing in places around these mechanisms. It's not clear why these vectors aren't symmetrical like the others. Generally the authors do a good job of exploring all possibilities allowed by the data, but in a few places this could be improved.

I think it would be useful to add a brief description of how one can get a change in δ¹³C without a change in CO₂ (e.g. the vertical lines, associated with sea ice), as this is what is happening during MIS4. It looks like this result is produced by certain combinations of NA and SO sea-ice change, but it's not

clear mechanistically why this happens, or how robust that result is across models. This would fit well around line 79.

Line 107: “The 72.1 ka feature is consistent with increased Southern Ocean gas exchange (Figure 2B). Southern ocean opal data show an upwelling event occurred at the onset of DO-19 (Figure 3), perhaps driven by a shift in the position or strength of the Southern Hemisphere Westerlies”. This text seems to imply that SO gas exchange and upwelling are the same thing. Yet in the cross plots they are clearly shown as different slopes. It would be useful to briefly discuss what is meant by SO gas exchange and how this would be accomplished - e.g. increased wind speed? Less sea-ice? If less sea-ice, why is the slope for gas exchange different to the one for SO sea-ice?

Line 174 further discusses sea-ice and gas exchange, and mentions sea-ice ‘modulates’ carbon export and gas exchange. This statement should be made directional- e.g. increased sea ice extent decreases carbon export and decreases gas exchange. Also, is it the case, in the model, that these two mechanisms cancel each other in terms of CO₂, which is why you get minimal change in CO₂ yet changes in carbon isotopes? This relates back to my query above about how you can get changes in carbon isotopes without changes in CO₂.

Line 176 – it’s worth mentioning here that in addition to timing problems, the cross-plots are also not consistent with changes in biological pump driving these changes in d¹³CO₂; there should have been changes in CO₂ along with the isotopes, so some compensating mechanism would be needed if the biological pump is invoked here.

Fig 2D- there is a rogue blue wedge here from panel E

Fig 3- It would be better to use the alkenone flux data and not concentration data. Similarly, it would be better to use the dust flux data from Lambert 2012, rather than the Ca concentration data.

Zanna Chase

Reviewer #2 (Remarks to the Author):

Menking et al. present new $\delta^{13}\text{C}\text{-CO}_2$ data from MIS 4, including the transitions before and after. The samples are from Taylor Glacier and therefore allow for a much higher resolution than previous data over this interval (Eggleston et al. 2016). Furthermore, this study extends the published record by Bauska et al. (2016) back in time.

The data were measured using established methods (Bauska et al. 2014) and similar analysis techniques to Bauska et al. (2016) and Eggleston et al. (2016). This is largely based on using Keeling plots based on results from two box models and two Earth system models of intermediate complexity. Although previous studies have assumed an approximate linear relationship among the several different mechanisms driving changes in CO_2 and $\delta^{13}\text{C}\text{-CO}_2$, I am a little concerned that that assumption may not be valid for the present study, as Menking and colleagues are not just trying to falsify a claim (i.e. that only a single process was responsible for the observed changes during MIS 4) but rather to estimate which processes may have caused these changes and to what extent each of these processes may have played a role. It would be very nice to see if one or more of the models do indeed produce linearly additive (and scalable) results, i.e. by adding different amounts of iron to the Southern Ocean or increasing the sea surface temperature by different amounts.

The use of mean ocean temperature measured on the same ice core is a strong indicator of the limitations of changes in alkalinity to affect $\delta^{13}\text{C}\text{-CO}_2$ over this period. However, this is a bit confusing to me, as alkalinity changes on long timescales. Isn't it possible that these data could show the imprint of a long-term trend due to alkalinity changes initiated by an event during MIS 5? Could the authors estimate how much this could realistically impact CO_2 over the course of MIS 4?

Figure S1 is very useful, but it seems to be missing some data; at least results regarding sea ice changes in the BICYCLE model! There, Köhler and Fischer (2006) found that changing the sea-ice cover in the northern (sink of atmospheric CO_2) and southern (source) hemispheres could have opposing effects on CO_2 . Have the authors considered this?

It would be interesting to know how quantitative the authors could be in assigning the changes in $\delta^{13}\text{C}\text{-CO}_2$ and CO_2 to the various mechanisms, as provided to some extent in figure S3. Would it be possible to at least provide statistics of the likelihood that various mechanisms were active during the different intervals identified here?

Minor comments:

page 3, line 9: The text states that "some models show rapid increase in CO_2 ", but only one model is referenced (Köhler et al. 2005). Could the authors provide at least one more reference to support this claim?

page 3, lines 14-15: The text states "The new MIS 5 data, better suited as analogues for today than previous studies of the last glacial period" is a little confusing; I would suggest adding "...for today than data from previous studies..." to clarify.

page 4, line 10: It seems odd that the two extreme values of $\delta^{13}\text{C}-\text{CO}_2$ given here do not have the same number of significant digits. Is there a reason for this?

Figure 2D: The blue stripe corresponding to "Lower sea surface temperature" seems to have run into panel C!

Figure 3: I don't believe the opal flux data by Anderson et al. (2009) are the most recent, although they probably sufficient for this figure. Sam Jaccard and/or Julia Gottschalk may have more recent (but possibly still unpublished) data.

Reviewer #3 (Remarks to the Author):

Review of manuscript submission "Multiple carbon cycle mechanisms associated with the glaciation of Marine Isotope Stage 4" by Menking et al.

Menking et al. present a detailed and thorough analysis of new and wonderfully high-precision CO_2 and $\delta^{13}\text{CO}_2$ measurements from Taylor Glacier covering the period between Marine Isotope Stage (MIS) 5a and the onset of MIS3, and discuss these in the light of paleoclimatological indicators from marine sediment cores and other climate archives. They are able to make a strong case that atmospheric CO_2 variability during that time interval was influenced by multiple processes that operated at the same time or in succession. The findings are in so far astonishing as no single process or sets of processes dominating the drawdown of CO_2 into the glaciation of MIS4 can be identified – something that is often underappreciated in the paleoclimatology community in my view. I therefore consider the contribution by Menking et al. as a highly important reminder of the complexity of the Earth system and those processes surrounding the initiation and termination of glacial periods. It becomes clear that a number of high-resolution paleoceanographic reconstructions are needed to better understand and verify the propositions made by Menking et al. I am hesitant as to whether the study's findings are sufficiently impactful to warrant publication in Nature Communications but the fact that I received it for review speaks against these doubts.

The manuscript is written in clear manner and the figures are of high quality. It is overall an excellent study and I have only minor comments that I hope will help the authors to improve their study.

I was a little bit confused by the structure of the manuscript as the paragraphs from line 43 to 69 read more like a Results section, which I find uncommon for a Nature Communications manuscript. I had difficulty to absorb all information in that section and remember them up to the point where the data were discussed later on. I would prefer a combination of results and discussion in this case. This would also alleviate the fact that a discussion of the data comes fairly late in the manuscript, essentially close to the half-way point of the manuscript.

I wished that the novelty of the major finding would be carved out a little bit better. The data suggest a complex succession of processes that influence CO₂ from MIS5a to the onset of MIS3 (e.g., line 10 and 46-47). I think it might be worth to highlight that these complexities are not reflected in the marine sedimentary record or in fact any other climate archive, are they?

In my view, some very interesting discussion is included in the supplementary text that would be equally suitable to be included in the main text. The strong link between CO₂ AND d¹³CO₂ changes during the onset of DO 19 that is absent during the onset of younger DO interstadials is an important finding that deserves more attention in the main text. Fig. S2 should in my view moved to the main text, and discussion of this should be expanded.

I was also puzzled at times about the use of "Southern Ocean data" because there is a fine line between different regions in the Southern Ocean, namely the subantarctic and Antarctic zone, that is not sufficiently acknowledged in the paper. The authors use paleoproductivity from both regions, and they in fact show very different processes, yet refer to them as "Southern Ocean data". For clarity, please specify the origin of the data from the Southern Ocean by stating "from the Antarctic Zone (core TNO57-14)" and "from the Subantarctic Zone (core xxx)" (e.g. line 140, Fig. 3) etc. Speaking of TNO57-14, it is important to note that Anderson et al. (2009) interpreted the opal flux peaks to occur during the stadial rather than the interstadial onset. Their age model is not good enough to resolve these fine nuances (i.e. whether the opal flux peak continues into DO19 or in fact covers only the stadial). I agree that the data are what they are, but maybe it is worthwhile to acknowledge these age uncertainties, in particular because the Anderson mechanism specifically refers to stadial conditions.

Line 14-16: how comes that Southern Ocean ventilation and sea ice changes did not impact CO₂?

Line 24: Often a single region is also highlighted that impacts on atmospheric CO₂. Maybe that is worth emphasizing too.

Line 28: insert “processes that led to a glacial drawdown of atmospheric CO₂ has been..”

Line 33: It would be good to name these intervals (x-y kyr BP) with their references for those who are not familiar with the existing data.

Line 37. Ocean circulation, also consider citing (Govin et al., 2009; Guihou et al., 2010)

Line 76. It is unclear how these categories were chosen. Do you think that they represent main drivers of CO₂ and δ¹³C_{CO₂ change? If yes, it would be good to clearly state that.}

Line 78-84. This paragraph seemingly provides possible scenarios of how a process changes CO₂ and δ¹³C_{CO₂. But it remains unclear why these specific scenarios were chosen. It remains unclear why for instance the influence of sea ice has not been considered here. This is not to say that these theoretical considerations are not useful. It is just unclear that emphasis is put on a CO₂ drop while the data clearly show a CO₂ increase during the late phase of MIS4.}

Line 100. Refer to Fig. 1.

Line 101. “these events”, better to say “DO events during MIS3” because I got confused here.

Line 102-104. Would the (Bereiter et al., 2012) mechanism help here to explain some of those differences?

Line 107. How does Southern Ocean gas exchange explain the 72.1 ka feature? A bit more explanation is required here. Fig. 2B also suggests that it could be explained through land carbon changes. Why is this not mentioned here?

Line 115. Maybe it is useful to consider (Matsumoto and Yokoyama, 2013) here?

Line 118. This question comes a little bit out of the blue because the readers might not immediately be aware of the data. I am aware that Fig. S2 has been referred to earlier but it would be useful to emphasize this topic more clearly in the main text. There should be room to do this.

Line 120-122. I must admit that I cannot follow this argument and the parallel that is drawn to present-day climate. This requires more explanation or should be removed. It is unclear how MIS5e would serve as an analog for today with much lower sea level and temperatures. The same goes with the final sentence of the main text in line 216-218. What positive climate-carbon feedback are here referred to?

Line 126. No new paragraph here.

Line 126-127. Reference missing.

Line 138. Remove “and” and specify where the benthic $\delta^{13}\text{C}$ records are from? I would also consider it key to plot the data by Jaccard et al. (2016) for instance in Fig. 3 as this backs up the authors hypothesis of a significant contribution of Southern Ocean carbon storage to atmospheric CO_2 changes. This record could replace the sea level record in Fig. 3 that is not really referred to in the text (apart from one instance where Fig. 3 is not even referenced).

Line 157. I would suggest no new paragraph here because this thematically goes with the previous paragraph.

Line 162. growing Southern Ocean sea ice could have

Line 163-165. What proxy data? Insert references.

Line 190. What proxy data? Insert references.

Line 192. Why is there such an emphasis on the Southern Ocean when it comes to air-sea gas exchange? How about the North Atlantic or North Pacific etc.

Line 198. end of Heinrich Stadial 6 would be better as termination is considered by some to mean something else.

Line 203. Concluding sentence here?

Line 209. It would also be good to highlight that some processes were active but did NOT alter CO₂. This is an important finding that should be emphasized more in my view.

Line 210. efficient nutrient utilization where?

Line 216. What kind of studies? I would make this a new sentence (after last glacial period).

Fig. 3. Should include the phases A, B.. from Fig. 1B here for clarity. It would also be helpful to add letters to each record and refer to these in the figure caption.

Line 360. Unclear why reference 78 was cited for the Southern Ocean opal flux.

References

Anderson, R.F., Ali, S., Bradtmiller, L.I., Nielsen, S.H.H., Fleisher, M.Q., Anderson, B., Burckle, L.H., 2009. Wind-driven upwelling in the Southern Ocean and the deglacial rise in atmospheric CO₂. *Science* (80-.). 323, 1443–1448. doi: 10.1126/science.1167441

Bereiter, B., Lüthi, D., Siegrist, M., Schüpbach, S., Stocker, T.F., Fischer, H., 2012. Mode change of millennial CO₂ variability during the last glacial cycle associated with a bipolar marine carbon seesaw. *Proc. Natl. Acad. Sci.* 109, 9755–9760. doi: 110.1073/pnas.1204069109

Govin, A., Michel, E., Labeyrie, L., Waelbroeck, C., Dewilde, F., Jansen, E., 2009. Evidence for northward expansion of Antarctic Bottom Water mass in the Southern Ocean during the last glacial inception. *Paleoceanography* 24, 1202. doi: 10.1029/2008PA001603

Guihou, A., Pichat, S., Nave, S., Govin, A., Labeyrie, L., Michel, E., Waelbroeck, C., 2010. Late slowdown of the Atlantic Meridional Overturning Circulation during the Last Glacial Inception: New constraints from sedimentary (231Pa/230Th). *Earth Planet. Sci. Lett.* 289, 520–529. doi: 10.1016/j.epsl.2009.11.045

Jaccard, S.L., Galbraith, E.D., Martínez-García, A., Anderson, R.F., 2016. Covariation of abyssal Southern Ocean oxygenation and pCO₂ throughout the last ice age. *Nature* 530, 207–210. doi: 10.1038/nature16514

Matsumoto, K., Yokoyama, Y., 2013. Atmospheric $\Delta^{14}\text{C}$ reduction in simulations of Atlantic overturning circulation shutdown. *Global Biogeochem. Cycles* 27, 296–304. doi: 10.1002/gbc.20035

REVIEWER COMMENTS

Reviewer #1 (Remarks to the Author):

This manuscript by Menking et al. presents new high resolution data on atmospheric CO₂ and δ¹³C-CO₂ concentrations across the last interglacial-glacial transition. The combination of these two proxies, together with the high resolution and precision of the data allow the authors to examine in detail the combination of carbon cycle mechanisms that were at play during this climate transition. The manuscript employs a clever model-data comparison approach in order to attribute mechanisms or combinations of mechanisms to different time periods within the overall climate transition. I also appreciate how it links these 'cross plot' interpretations against multi-proxy datasets (fig 3).

While the manuscript raises a lot of questions that it cannot answer, it is an important contribution as it shows how multiple mechanisms can act together to produce transitions in CO₂, and how these mechanisms can be different depending on the 'starting' conditions. The possibility that carbon feedbacks may be stronger than we thought, based on MIS 3 analogs, is an important message, and shows the carbon-climate relationship still has surprises in store. The dataset and hypotheses put forward will stimulate future research in this area.

The manuscript is well written and the figures are informative. I don't have any major concerns, just a number of suggestions and queries to improve the manuscript.

We thank Dr. Zanna Chase for a thoughtful and supportive review. We have addressed the comments below and noted where specific changes were made in the manuscript.

I think the biggest weakness of this study is the reliance on a compilation of model runs, which are not exactly the same in terms of their set up or perturbations. While arguably this makes the results more robust than using a single model, it does complicate the interpretation, and an argument could be made that it would be better to use a single model with a range of sensitivities.

We appreciate the suggestion that using a single model would be a better approach. The paper was originally written only using results from the OSU box model, but additional model results were added at a later stage to compare to the results from the OSU box model. In our view, the interpretations still rely dominantly on the OSU model, and the additional models (UVic, Bicycle, Bern3d, and LOVECLIM) are used to show general agreement with and therefore support the results from the OSU model.

Despite the models having different setups, sensitivities, and experimental designs, the results of similar perturbations tend to plot with similar slopes. For example, the biological pump was perturbed by all four of the models, but in different ways (changes in wind speed, AMOC, and productivity), and the resulting changes in δ¹³C-CO₂ and CO₂ have similar slopes in the cross plots. So the δ¹³C-CO₂ and CO₂ data can identify certain types of carbon cycle changes (e.g., biological pump changes) and constrain the problem to a narrower set of processes. It is our view that the compilation is beneficial for setting up a heuristic framework, and it demonstrates that our interpretations are not dependent on a single model. We also note that Reviewer 2 states that the model compilation is "extremely useful" and requested that more model data be added.

We were prompted by the comment to revise the presentation of the modelling work, but ultimately keep the compilation. We now show in the supplement Figure S1: (1) results of OSU box model experiments re-printed from Bauska et al. 2016 including results of new modelling experiments testing (at reviewer's request) perturbations of opposite magnitude (such as cooling SST rather than warming), (2) tests demonstrating the effect on δ¹³C-CO₂ and CO₂ of changes in

northern and southern sea ice extent that result in zero net CO₂ change, and (3) tests demonstrating the linear scalability of the box model results. In Figure S2 we show how results from the Bicycle, UVic, Bern3d, and LOVECLIM models compare to the OSU model results.

The biological pump and SST perturbations are fairly straightforward, but there is a lot of overlap in the land carbon/sea ice/gas exchange mechanisms, and the text is confusing in places around these mechanisms. It's not clear why these vectors aren't symmetrical like the others. Generally the authors do a good job of exploring all possibilities allowed by the data, but in a few places this could be improved.

We agree it is confusing that not all results were displayed symmetrically. We revised the presentation of the modeling in the supplement to show all results symmetrically, except for land carbon uptake, which we justify is not plotted because it is not likely to be a cause of CO₂ drop at the MIS 5-4 transition or any other interval in our study (supplementary material lines 82-84).

Since the reviewer did not specifically say where in the text we might improve "exploring all possibilities allowed by the data," it is difficult to know what else to revise.

I think it would be useful to add a brief description of how one can get a change in d13C without a change in CO₂ (e.g. the vertical lines, associated with sea ice), as this is what is happening during MIS4. It looks like this result is produced by certain combinations of NA and SO sea-ice change, but it's not clear mechanistically why this happens, or how robust that result is across models. This would fit well around line 79.

We revised text in lines 91-94 and lines 218-224 and lines 231-235 to explain the model result of extending Antarctic and North Atlantic sea ice simultaneously to get a change in d¹³C-CO₂ with no (or very little) accompanying change in CO₂. We included a model run that specifically demonstrates this (Figure S1). Following on comments from Reviewer 2, we also refer to model results from Kohler 2006 that demonstrate changes in sea ice for the extremes of full shrinkage of N. Atl. and Southern Ocean sea ice from 100% coverage to preindustrial values with compensating effects on CO₂.

Line 107: "The 72.1 ka feature is consistent with increased Southern Ocean gas exchange (Figure 2B). Southern ocean opal data show an upwelling event occurred at the onset of DO-19 (Figure 3), perhaps driven by a shift in the position or strength of the Southern Hemisphere Westerlies". This text seems to imply that SO gas exchange and upwelling are the same thing. Yet in the cross plots they are clearly shown as different slopes. It would be useful to briefly discuss what is meant by SO gas exchange and how this would be accomplished - e.g. increased wind speed? Less sea-ice? If less sea-ice, why is the slope for gas exchange different to the one for SO sea-ice?

We note the ambiguity in the original text, and we added text clarifying what we mean by SO gas exchange. We agree that upwelling and gas exchange are not the same thing. S.O. gas exchange could be increased by increasing wind speed or decreasing sea ice. Upwelling could be increased by increasing wind speed. Therefore, upwelling and S.O. gas exchange might both be driven by a shift in the strength or position of the Westerlies. It is reasonable to think, then, that increased S.O. gas exchange rates would accompany increased S.O. upwelling. The link between gas exchange and westerlies is now noted in lines 131-133 and lines 222-224. We also revised Figure S1 to show the results of Antarctic sea ice changes, and adjusted the upwards purple region representing SO gas exchange so it overlaps with the sea ice changes. Decreasing the gas exchange parameter produces results that look nearly identical to the Antarctic sea ice increases (not shown). The simplicity of the ocean circulation in the OSU box model and lack of dynamic atmosphere prevent us from driving SO

“upwelling” with westerlies wind shifts, so we changed wording in the main text from “ventilation” to specifically say “air-sea gas exchange” to avoid any confusion or ambiguity around this mechanism. This change was made on line 14, 17, 20, and 271.

Line 174 further discusses sea-ice and gas exchange, and mentions sea-ice ‘modulates’ carbon export and gas exchange. This statement should be made directional- e.g. increased sea ice extent decreases carbon export and decreases gas exchange. Also, is it the case, in the model, that these two mechanisms cancel each other in terms of CO₂, which is why you get minimal change in CO₂ yet changes in carbon isotopes? This relates back to my query above about how you can get changes in carbon isotopes without changes in CO₂.

We added text to supplementary lines 41-44 specify increasing sea ice decreases carbon export and decreases gas exchange.

Increasing Antarctic sea ice alone does not affect primary production enough to cancel the CO₂ changes, hence extension of Ant. sea ice results in the vector in Figure S1 with -17 ppm change to CO₂. You would need a second compensating process (like N. Atl. sea ice extension or bio pump decrease in the subantarctic) to contribute in tandem with Ant. sea ice.

Line 176 – it’s worth mentioning here that in addition to timing problems, the cross-plots are also not consistent with changes in biological pump driving these changes in d13CO₂; there should have been changes in CO₂ along with the isotopes, so some compensating mechanism would be needed if the biological pump is invoked here.

We added text to lines 229-231 to read, “but the cross-plots suggest there must have been some compensating mechanism to offset changes in CO₂ if the biological pump is invoked (Figure 2E).”

Fig 2D- there is a rogue blue wedge here from panel E

This is fixed.

Fig 3- It would be better to use the alkenone flux data and not concentration data. Similarly, it would be better to use the dust flux data from Lambert 2012, rather than the Ca concentration data.

We have plotted the dust flux data from EDC (Lambert et al. 2012) and the alkenone flux data from TN057-21 (Anderson et al. 2014) instead of the concentration data.

Zanna Chase

Reviewer #2 (Remarks to the Author):

Menking et al. present new $\delta^{13}\text{C-CO}_2$ data from MIS 4, including the transitions before and after. The samples are from Taylor Glacier and therefore allow for a much higher resolution than previous data over this interval (Eggleston et al. 2016). Furthermore, this study extends the published record by Bauska et al. (2016) back in time.

We thank Reviewer #2 for helpful comments about the paper. Below we respond to each comment noting where we have revised the text accordingly.

The data were measured using established methods (Bauska et al. 2014) and similar analysis techniques to Bauska et al. (2016) and Eggleston et al. (2016). This is largely based on using Keeling plots based on results from two box models and two Earth system models of intermediate complexity. Although previous studies have assumed an approximate linear relationship among the several different mechanisms driving changes in CO_2 and $\delta^{13}\text{C-CO}_2$, I am a little concerned that that assumption may not be valid for the present study, as Menking and colleagues are not just trying to falsify a claim (i.e. that only a single process was responsible for the observed changes during MIS 4) but rather to estimate which processes may have caused these changes and to what extent each of these processes may have played a role. It would be very nice to see if one of more of the models do indeed produce linearly additive (and scalable) results, i.e. by adding different amounts of iron to the Southern Ocean or increasing the sea surface temperature by different amounts.

We revised how the modeling is presented in the supplementary material, including a short section that demonstrates how the model results are linearly additive and scalable using the OSU box model. These results are now shown in Figure S1.

The use of mean ocean temperature measured on the same ice core is a strong indicator of the limitations of changes in alkalinity to affect $\delta^{13}\text{C-CO}_2$ over this period. However, this is a bit confusing to me, as alkalinity changes on long timescales. Isn't it possible that these data could show the imprint of a long-term trend due to alkalinity changes initiated by an event during MIS 5? Could the authors estimate how much this could realistically impact CO_2 over the course of MIS 4?

We are slightly confused by this comment, because we did not claim that “mean ocean temperature measured on the same ice core is a strong indicator of the limitations of changes in alkalinity to affect $\delta^{13}\text{C-CO}_2$ over this period.” We suspect Reviewer 2 is referring to the statement from Page 2 Lines 97-100, “However, we can effectively remove two degrees of freedom in the system by employing co-eval constraints on mean ocean temperature provided by noble gas measurements, and ruling out any changes due to the slow response of the CaCO_3 cycle (e.g. weathering, reef building, dissolution/burial) for rapid variations in CO_2 .” The intended meaning is: “We can effectively remove two degrees of freedom in the system: (1) Noble gas measurements allow co-eval constraints on mean ocean temperature changes, and (separately) (2) we can rule out adjustment of the CaCO_3 cycle (at least for fast variations in CO_2) because they are too slow. We revised the text in lines 103-106 accordingly.

Carbonate compensation could be responsible for a small amount of CO_2 variation during the more gradual CO_2 change during the descent into MIS 4. It would be evident as change in CO_2 with no or very little change in $\delta^{13}\text{C-CO}_2$. We do not see intervals like this in our dataset, rather the data tend to show large and rapid changes in $\delta^{13}\text{C-CO}_2$. For this reason, we did not think CaCO_3 was very important for explaining the data, however it could be that some slow CO_2 change due to CaCO_3

compensation from a prior event is masked underneath the larger changes. It is difficult to estimate since we do not have isotope data spanning MIS 5 prior to 74.1 ka, but we can make a modest estimate of the effect from, say, the CO₂ variation at DO 20 or DO 19 due to land carbon release. We conservatively estimate the CO₂ drawdown could not be attributed to more than a gradual 4 ppm decrease in CO₂ concentration, if imbalance of ocean carbonate chemistry with marine sediments ensued following DO-20 and DO-19. We added lines 191-197 to the main text, “We also rule out CaCO₃ compensation as a significant player in the CO₂ drawdown because (1) CaCO₃ compensation operates on a multi-millennial timescale, and (2) the predicted pattern of CO₂ and δ¹³C-CO₂ changes (change in CO₂ accompanied by little to no change in δ¹³C-CO₂) is not evident in the data (Figure 2D). If CO₂ decrease was due to slow CaCO₃ compensation in response to prior events that occurred during MIS 5a, it would have been masked by larger carbon cycle changes during the MIS 5-4 transition and was not likely to contribute more than 4 ppm to the CO₂ decline,” and supplementary lines 213-222 “We also suggest that CaCO₃ compensation played a minimal role in the CO₂ drawdown. We estimate that CaCO₃ compensation in response to a pulse of land carbon at DO-19 could have contributed up to 3 ppm to the decrease in CO₂ over the following three thousand years after DO-19. We do not have data to constrain carbon cycle changes prior to 74.1 ka (e.g., DO-20 and the transition to stadial conditions preceding DO-19), but conservatively, we estimate continued adjustment of the marine carbonate system with CaCO₃ sediments in response to an event at DO-20 could have conspired with adjustments following DO-19 to cause a maximum 5 ppm reduction in CO₂ attributed to CaCO₃ compensation. This process would have occurred gradually over the course of the full MIS 5-4 transition and would do nothing to explain the large δ¹³C-CO₂ decrease and rebound centered at 70.5 ka.”

Figure S1 is very useful, but it seems to be missing some data; at least results regarding sea ice changes in the BICYCLE model! There, Köhler and Fischer (2006) found that changing the sea-ice cover in the northern (sink of atmospheric CO₂) and southern (source) hemispheres could have opposing effects on CO₂. Have the authors considered this?

We thank Reviewer 2 for pointing out the Kohler and Fischer (2006) result. We have considered this, and it is the reason why the blue “extend sea ice” region in Figure S1 covers such a broad swath of space in the figure. The exact δ¹³C-CO₂-to-CO₂ relationship that results from sea ice extension depends on the fraction of sea ice extension in the north versus the south, as they have compensating effects. It is possible to extend sea ice in both hemispheres (as was likely the case during the descent to MIS 4) such that there is a cancelling effect on CO₂, but still a net increase or decrease in δ¹³C-CO₂. Reviewer 1 also asked about this point, and requested a bit more explanation about how one might achieve a change in δ¹³C-CO₂ with no net change in CO₂ concentration. We reference this result in lines 91-94 and lines 221-224, as well as a small amount of discussion in lines 231-235. In the supplementary material we added lines 41-57 to discuss the sea ice results in the box model including reference to Kohler 2006. We also plotted the extreme N. Atlantic and Southern Ocean sea ice results from (Kohler et al. 2006), now in Figure S2).

It would be interesting to know how quantitative the authors could be in assigning the changes in δ¹³C-CO₂ and CO₂ to the various mechanisms, as provided to some extent in figure S3. Would it be possible to at least provide statistics of the likelihood that various mechanisms were active during the different intervals identified here?

We extensively considered this during the preparation of the manuscript. Quantitative determination of the changes in the CO₂ sources/ sinks (often referred to in the literature as single- or double- deconvolution) is not mathematically possible given the number of variables operating in the carbon cycle across large transitions like the MIS 5-4 or the MIS 4-3. In order to do this kind of analysis, one would have to remove variables either with detailed paleoclimate data (e.g., fixed SST, land carbon, sea ice histories), or by making large assumptions about the histories of certain variables (e.g., assume no change, assume hypothetical histories).

We would prefer to present the data in this paper without doing the suggested analysis for several reasons. For one, applying the deconvolution analysis to intervals like the last glacial period is still a relatively new application of the tool, and we would be pushing the limits of the approach given the sparsity of paleoclimate data to help add constraints. We prefer to let this paper highlight the data, be up-front about the limitations of $d^{13}\text{C-CO}_2$ and CO_2 , and to allow future, more extensive modelling studies address the hypotheses we propose.

We did take the reviewers suggestion to expand on the original supplementary figure S3. We now show forward simulations of all four intervals highlighted in the main text. The forward simulations attempt to show how processes might work in combination to influence CO_2 and $d^{13}\text{C-CO}_2$ in different ways. This work is now presented in supplementary figures S3-S6 and associated text.

Minor comments:

page 3, line 9: The text states that "some models show rapid increase in CO_2 ", but only one model is referenced (Köhler et al. 2005). Could the authors provide at least one more reference to support this claim?

We changed the wording in line 143 to reflect that it is only based on one study.

page 3, lines 14-15: The text states "The new MIS 5 data, better suited as analogues for today than previous studies of the last glacial period" is a little confusing; I would suggest adding "...for today than data from previous studies..." to clarify.

We changed the wording on line 151 as suggested.

page 4, line 10: It seems odd that the two extreme values of $d^{13}\text{C-CO}_2$ given here do not have the same number of significant digits. Is there a reason for this?

We changed the numbers on lines 55-57 to consistently have two significant digits.

Figure 2D: The blue stripe corresponding to "Lower sea surface temperature" seems to have run into panel C!

This is fixed!

Figure 3: I don't believe the opal flux data by Anderson et al. (2009) are the most recent, although they probably sufficient for this figure. Sam Jaccard and/or Julia Gottschalk may have more recent (but possibly still unpublished) data.

We agree the data are sufficient for this figure.

Reviewer #3 (Remarks to the Author):

Review of manuscript submission “Multiple carbon cycle mechanisms associated with the glaciation of Marine Isotope Stage 4” by Menking et al.

Menking et al. present a detailed and thorough analysis of new and wonderfully high-precision CO₂ and δ¹³C-CO₂ measurements from Taylor Glacier covering the period between Marine Isotope Stage (MIS) 5a and the onset of MIS3, and discuss these in the light of paleoclimatological indicators from marine sediment cores and other climate archives. They are able to make a strong case that atmospheric CO₂ variability during that time interval was influenced by multiple processes that operated at the same time or in succession. The findings are in so far astonishing as no single process or sets of processes dominating the drawdown of CO₂ into the glaciation of MIS4 can be identified – something that is often underappreciated in the paleoclimatology community in my view. I therefore consider the contribution by Menking et al. as a highly important reminder of the complexity of the Earth system and those processes surrounding the initiation and termination of glacial periods. It becomes clear that a number of high-resolution paleoceanographic reconstructions are needed to better understand and verify the propositions made by Menking et al. I am hesitant as to whether the study’s findings are sufficiently impactful to warrant publication in Nature Communications but the fact that I received it for review speaks against these doubts.

The manuscript is written in clear manner and the figures are of high quality. It is overall an excellent study and I have only minor comments that I hope will help the authors to improve their study.

We thank Reviewer 3 for a supportive and insightful review. Please see below for responses and specific revisions to the manuscript.

I was a little bit confused by the structure of the manuscript as the paragraphs from line 43 to 69 read more like a Results section, which I find uncommon for a Nature Communications manuscript. I had difficulty to absorb all information in that section and remember them up to the point where the data were discussed later on. I would prefer a combination of results and discussion in this case. This would also alleviate the fact that a discussion of the data comes fairly late in the manuscript, essentially close to the half-way point of the manuscript.

Because the data represent a significant improvement in resolution and precision in atmospheric δ¹³C-CO₂ for this time period, and because the data illustrate not-before-seen and surprising changes in δ¹³C-CO₂, we prefer to keep the section the reviewer is referring to. We will defer to the editor’s decision on this if reorganization of the text is preferred.

I wished that the novelty of the major finding would be carved out a little bit better. The data suggest a complex succession of processes that influence CO₂ from MIS5a to the onset of MIS3 (e.g., line 10 and 46-47). I think it might be worth to highlight that these complexities are not reflected in the marine sedimentary record or in fact any other climate archive, are they?

To a large extent, we already did discuss how the new data are reflected in other paleoceanographic data. Examples of this are: lines 133-137, lines 170-177, lines 225-231. We added the text, “The δ¹³C-CO₂ anomaly is a difficult feature to explain, and is not reflected in other paleoceanographic data (Figure 4)” to lines 182-183 to point out what the reviewer suggested. We also added lines 250-255: “It is worth emphasizing that the feature represents the largest magnitude decrease in δ¹³C-CO₂ in the ice core record, exceeded only in magnitude by the decrease in δ¹³C-CO₂ observed during the industrial era due to the

combustion of fossil fuels. It is also notable that the changes observed in other paleoceanographic records between 63.6-60 ka mostly do not reflect the exceptional nature of the $\delta^{13}\text{C}$ -CO₂ change (Figure 4), which makes the $\delta^{13}\text{C}$ -CO₂ feature difficult to explain.”

In my view, some very interesting discussion is included in the supplementary text that would be equally suitable to be included in the main text. The strong link between CO₂ AND d13CO₂ changes during the onset of DO 19 that is absent during the onset of younger DO interstadials is an important finding that deserves more attention in the main text. Fig. S2 should in my view moved to the main text, and discussion of this should be expanded.

We moved the figure to the main text, now Figure 3.

I was also puzzled at times about the use of “Southern Ocean data” because there is a fine line between different regions in the Southern Ocean, namely the subantarctic and Antarctic zone, that is not sufficiently acknowledged in the paper. The authors use paleoproductivity from both regions, and they in fact show very different processes, yet refer to them as “Southern Ocean data”. For clarity, please specify the origin of the data from the Southern Ocean by stating “from the Antarctic Zone (core TNO57-14)” and “from the Subantarctic Zone (core xxx)” (e.g. line 140, Fig. 3) etc. Speaking of TN057-14, it is important to note that Anderson et al. (2009) interpreted the opal flux peaks to occur during the stadial rather than the interstadial onset. Their age model is not good enough to resolve these fine nuances (i.e. whether the opal flux peak continues into DO19 or in fact covers only the stadial). I agree that the data are what they are, but maybe it is worthwhile to acknowledge these age uncertainties, in particular because the Anderson mechanism specifically refers to stadial conditions.

We changed the text in lines 173-175 to read, “Proxies for productivity and iron fertilization show that organic carbon export increased in the Subantarctic South Atlantic” in order to distinguish from the Antarctic region south of the polar front. We specified “Subantarctic ocean biological productivity” to line 201-202. We also added the specific core names to Figure 4 caption. We note that our d13C-CO₂ and CO₂ data cannot distinguish the region of the ocean where productivity changes occurred, and in some ways, it is unimportant to our interpretations, but we agree that in the text we should be specific where appropriate, such as when referring to other authors’ work. We noted in the main text that the Anderson mechanism refers to stadial conditions, and we use this as a counterpoint for the DO-19 change being driven by SO gas exchange (lines 137-138).

Line 14-16: how comes that Southern Ocean ventilation and sea ice changes did not impact CO₂?

The data indicate that CO₂ did not change during Mis 4 very much, despite d13C-CO₂ changing substantially. It implies that there are either processes active in the carbon cycle that do not change CO₂, or there are multiple processes active with compensating effects on CO₂. We highlighted this more deliberately in the text lines 217-236 and in the supplementary material lines 41-57, also in response to comments by other reviewers.

Line 24: Often a single region is also highlighted that impacts on atmospheric CO₂. Maybe that is worth emphasizing too.

We added “or a single region.” to Line 27-28

Line 28: insert “processes that led to a glacial drawdown of atmospheric CO₂ has been..”

We changed the text accordingly in line 28.

Line 33: It would be good to name these intervals (x-y kyr BP) with their references for those who are not familiar with the existing data.

We added ages for those who are not familiar with the intervals we named in the abstract as well as lines 34-37.

Line 37. Ocean circulation, also consider citing (Govin et al., 2009; Guihou et al., 2010)

We have added these references to Line 41.

Line 76. It is unclear how these categories were chosen. Do you think that they represent main drivers of CO₂ and δ¹³C_{CO₂} change? If yes, it would be good to clearly state that.

We added, “representing the primary drivers of glacial-interglacial and millennial-scale CO₂ change discussed in the literature ^{1,2}” to lines 85-86.

Line 78-84. This paragraph seemingly provides possible scenarios of how a process changes CO₂ and δ¹³C_{CO₂}. But it remains unclear why these specific scenarios were chosen. It remains unclear why for instance the influence of sea ice has not been considered here. This is not to say that these theoretical considerations are not useful. It is just unclear that emphasis is put on a CO₂ drop while the data clearly show a CO₂ increase during the late phase of MIS4.

This paragraph was meant simply to illustrate some of the groups of results on the cross-plot. We added text about sea ice, specifically about large changes in δ¹³C-CO₂ with no accompanying CO₂ change (as requested by other reviewers) to lines 91-94. The emphasis on a CO₂ drop is largely based on the motivation posed in the introduction where we stated “Processes that led to a glacial drawdown of CO₂ received less attention, despite their importance for understanding glacial-interglacial CO₂ cycles.”

Line 100. Refer to Fig. 1.

We do not think a reference to Figure 1 helps illustrate this point.

Line 101. “these events”, better to say “DO events during MIS3” because I got confused here.

We said “MIS 3 DO events” on line 124.

Line 102-104. Would the (Bereiter et al., 2012) mechanism help here to explain some of those differences?

Our new data do not refute or bolster Bereiter’s hypothesis, which was that CO₂ during MIS 3 DO events was different because of a glacial state ocean. Our CO₂ at DO 19 fits with the Bereiter observation in that there is not a significant lag of CO₂ peak behind the onset of the DO warming,

but we add new information – the $\delta^{13}\text{C}-\text{CO}_2$ – which shows a significant depletion, pointing to land carbon or increased Southern Ocean gas exchange as the mechanism. Our point in this paragraph is that the DO19 $\delta^{13}\text{C}-\text{CO}_2$ data are distinct from the limited $\delta^{13}\text{C}-\text{CO}_2$ data from MIS 3, which do not show the negative excursion. We added this text to lines 118-119, “Previous work highlighted a contrast between millennial-scale CO_2 changes during MIS 5 versus MIS 3, with local maxima in CO_2 occurring closer to the onset of DO events during MIS 5³.” to reference Bereiter 2012 and contrast with our discussion, but note that the Bereiter idea of a greater lag in CO_2 maxima during MIS 3 is not precisely relevant because we are contrasting the fast DO-like CO_2 changes in MIS 3 with the CO_2 change at DO-19, specifically the isotope trends.

Line 107. How does Southern Ocean gas exchange explain the 72.1 ka feature? A bit more explanation is required here. Fig. 2B also suggests that it could be explained through land carbon changes. Why is this not mentioned here?

The statement refers to the slope of the data being similar to the slope of the model results. We changed the wording to read, “The cross-plot suggests the changes in $\delta^{13}\text{C}-\text{CO}_2$ and CO_2 across DO-19 (72.1 ka) are consistent with increased Southern Ocean gas exchange (Figure 2C), which could have resulted from a shift in the strength or position of the Southern Hemisphere Westerlies⁴.” The land carbon explanation is discussed afterwards in lines 140-153.

Line 115. Maybe it is useful to consider (Matsumoto and Yokoyama, 2013) here?

The referenced work is about deep ocean ventilation and reduction in atmospheric $\Delta^{14}\text{C}$ due to ventilation of the deep Southern Ocean in response to freshwater forcing of an AMOC shutdown. We would expect this process to be relevant during the stadial preceding DO-19, but in this paragraph, we are discussing the rapid CO_2 jump and negative isotopic trend that occurs in phase with the DO-19 warming. It is a different mode of CO_2 variability than what Matsumoto and Yokoyama 2013 modeled.

Line 118. This question comes a little bit out of the blue because the readers might not immediately be aware of the data. I am aware that Fig. S2 has been referred to earlier but it would be useful to emphasize this topic more clearly in the main text. There should be room to do this.

We changed the wording in lines 145-150 to not be phrased as a question. It now reads, “The rapid CO_2 increase and large $\delta^{13}\text{C}-\text{CO}_2$ decrease at DO-19 may be an example of this mode of change, but the stark contrast in $\delta^{13}\text{C}-\text{CO}_2$ across DO-19 versus DO-8 (Figure 3) would imply that the response of the terrestrial biosphere to AMOC recovery was different at different times. One reason might be that the terrestrial biosphere was larger during MIS 5a relative to MIS 3 and therefore sudden changes in Northern Hemisphere temperature had a greater effect on the atmospheric CO_2 budget.”

Line 120-122. I must admit that I cannot follow this argument and the parallel that is drawn to present-day climate. This requires more explanation or should be removed. It is unclear how MIS5e would serve as an analog for today with much lower sea level and temperatures. The same goes with the final sentence of the main text in line 216-218. What positive climate-carbon feedback are here referred to?

The carbon-climate feedback referred to is the feedback between land carbon and temperature. The DO 19 data may suggest the terrestrial biosphere is more sensitive to warming during MIS 5 than during MIS 3. The suggestion is based on our observation that the negative isotope excursion we see

at DO 19 is not apparent at DO 8 (Bauska 2018). We suggest that, while of course not a perfect analog, MIS 5a is perhaps a better analog for today than MIS 3, because it was warmer and ice sheets were smaller.

Line 126. No new paragraph here.

We merged the two paragraphs.

Line 126-127. Reference missing.

We added Bereiter 2012 to line 157.

Line 138. Remove “and” and specify where the benthic d13C records are from? I would also consider it key to plot the data by Jaccard et al. (2016) for instance in Fig. 3 as this backs up the authors hypothesis of a significant contribution of Southern Ocean carbon storage to atmospheric CO₂ changes. This record could replace the sea level record in Fig. 3 that is not really referred to in the text (apart from one instance where Fig. 3 is not even referenced).

We plotted the Jaccard 2016 data instead of relative sea level. We also added the reference to “(Figure 4)” on line 170-173 and stated where the benthic d13C come from (the South Atlantic).

Line 157. I would suggest no new paragraph here because this thematically goes with the previous paragraph.

We combined the paragraphs.

Line 162. growing Southern Ocean sea ice could have

We changed the wording in line 203-205 to “Increases in the efficiency of the biological pump and Antarctic sea ice coverage could have sequestered CO₂ in the deep ocean, which lowered CO₂ and raised δ¹³C-CO₂ beyond the values prior to the excursion.”

Line 163-165. What proxy data? Insert references.

We meant that the proxy data in Figure 4 show this. We referenced these datasets specifically in line 180-181 (Wolff 2006, Jaccard 2016, Thornalley 2013, Anderson 2014, Yu 2016).

Line 190. What proxy data? Insert references.

Similar to above, we referenced (Jaccard 2016, Thornalley 2013, Anderson 2014, Yu 2016) in line 205-207.

Line 192. Why is there such an emphasis on the Southern Ocean when it comes to air-sea gas exchange? How about the North Atlantic or North Pacific etc.

The new d13C-CO₂ and CO₂ data show many instances of large, negative and positive isotopic changes accompanying relatively small CO₂ changes. In the model compilation, increasing/decreasing gas exchange rate in the Southern Ocean is one of the few processes (along with fast land carbon release) that achieves such high slopes in the cross-plots. We focused on the Southern Ocean

for this reason, as well as because of the predominance of the Southern Ocean and Westerlies wind shifts in the glacial-interglacial CO₂ literature.

Line 198. end of Heinrich Stadial 6 would be better as termination is considered by some to mean something else.

We replaced “termination” with “end” on line 265.

Line 203. Concluding sentence here?

We concluded with: “The mechanisms invoked to explain the CO₂ rise across the MIS 4-3 transition are not unlike those that explain the rise in CO₂ across the last deglacial transition ^{5,6}. One key difference between the two intervals, and a plausible explanation for why the MIS 4-3 change in δ¹³C-CO₂ was so great, is that the carbon cycle changes were less convolved with the impact of rising sea surface temperature. Ocean heating is estimated to have contributed only ~ 10 ppm to the CO₂ rise at the MIS 4-3 transition ⁷, but ~ 30 ppm during the last deglaciation, which would partially cancel the impact on δ¹³C-CO₂ of a relaxed biological pump or enhanced Southern Ocean gas exchange.”

Line 209. It would also be good to highlight that some processes were active but did NOT alter CO₂. This is an important finding that should be emphasized more in my view.

We added, “The data also demonstrate that processes were active during MIS 4 that altered δ¹³C-CO₂ with little to no change in CO₂ concentration” lines 288-290.

Line 210. efficient nutrient utilization where?

We changed “more efficient nutrient utilization” to read “more efficient biological pump,” and the meaning is globally. (line 285)

Line 216. What kind of studies? I would make this a new sentence (after last glacial period).

We changed the wording of lines 293-294 to read, “The result may suggest that a previous study which concluded positive climate-carbon feedbacks were small during abrupt warmings needs further examination using better suited climatic analogs ⁸.”

Fig. 3. Should include the phases A, B.. from Fig. 1B here for clarity. It would also be helpful to add letters to each record and refer to these in the figure caption.

We added the phases I, II, III, IV from Figure 1B and Figure 2C-F. We added letters to each record and refer to them in the figure caption and main text. We also changed the interval labels in Figure 1B, Figure 2 C-F, and Figure 4 from letters to Roman numerals, i.e. to “I, II, III, IV” to not confuse them with the new letters in Figure 4. We updated all references in the text accordingly.

Line 360. Unclear why reference 78 was cited for the Southern Ocean opal flux.

We deleted this reference.

- 1 Gottschalk, J. *et al.* Mechanisms of millennial-scale atmospheric CO₂ change in numerical model simulations. *Quat. Sci. Rev.* **220**, 30-74, doi:10.1016/j.quascirev.2019.05.013 (2019).
- 2 Sigman, D. M. & Boyle, E. A. Glacial/interglacial variations in atmospheric carbon dioxide. *Nature* **407**, 859-869, doi:10.1038/35038000 (2000).
- 3 Bereiter, B. *et al.* Mode change of millennial CO₂ variability during the last glacial cycle associated with a bipolar marine carbon seesaw. *Proceedings of the National Academy of Sciences of the United States of America* **109**, 9755-9760, doi:10.1073/pnas.1204069109 (2012).
- 4 Tschumi, T., Joos, F. & Parekh, P. How important are Southern Hemisphere wind changes for low glacial carbon dioxide? A model study. *Paleoceanography* **23**, 20, doi:10.1029/2008pa001592 (2008).
- 5 Bauska, T. *et al.* Carbon isotopes characterize rapid changes in atmospheric carbon dioxide during the last deglaciation. *Proceedings of the National Academy of Sciences of the United States of America* **113**, 3465-3470, doi:10.1073/pnas.1513868113 (2016).
- 6 Schmitt, J. *et al.* Carbon Isotope Constraints on the Deglacial CO₂ Rise from Ice Cores. *Science* **336**, 711-714, doi:10.1126/science.1217161 (2012).
- 7 Shackleton, S. *et al.* Evolution of mean ocean temperature in Marine Isotope Stage 4. *Climate of the Past* **17**, 2273-2289 (2021).
- 8 Bauska, T. K. *et al.* Controls on Millennial-Scale Atmospheric CO₂ Variability During the Last Glacial Period. *Geophysical Research Letters* **45**, 7731-7740, doi:10.1029/2018gl077881 (2018).

REVIEWERS' COMMENTS

Reviewer #2 (Remarks to the Author):

Thanks very much for the revised article! I only see a few remaining points that would be great if you could address/clarify:

1. In the supplement (line 83), you write that carbon uptake by the terrestrial biosphere does not seem to be relevant during the period of this study. However, this mechanism is mentioned at least in the manuscript in lines 133-134, so it seems like it could be useful to include this in figure 2 (and figures S1 and S2) - if it doesn't make the figures too messy and difficult to understand.
2. Figure S2 seems to be missing a label that probably corresponds to a decrease in wind speed in the Bern 3D model.
3. I realize that the shaded regions in the cross plots are taken from Bauska et al. (2016), but it's unclear to me how the ranges were calculated (I apologize if I missed this!). Furthermore, not all of the models used in this study fall into these shaded regions (figure S2). Could you expound upon this somewhere?
4. Please double-check that you have appropriately cited all external datasets that you're plotting in figures 1, 3, and 4. I believe there's a reference missing to Eggleston et al. (2015) in figure 1A, for example.

Reviewer #3 (Remarks to the Author):

Rereview of Menking et al. Multiple carbon cycle mechanisms associated with the glaciation of Marine Isotope Stage 4 (Submission NCOMMS-21-51287A)

I have read the revised submission of Menking et al., and their response to my previous comments and those of the other two reviewers. In my view the authors have responded efficiently and satisfactorily to all points of criticism. I congratulate the authors for their effective revision of the study and for the study itself that will be an important contribution once published and spark new research on the mechanisms driving (the transition in and out of) MIS4 by both, the proxy and modelling community.

REVIEWERS' COMMENTS

Reviewer #2 (Remarks to the Author):

Thanks very much for the revised article! I only see a few remaining points that would be great if you could address/clarify:

1. In the supplement (line 83), you write that carbon uptake by the terrestrial biosphere does not seem to be relevant during the period of this study. However, this mechanism is mentioned at least in the manuscript in lines 133-134, so it seems like it could be useful to include this in figure 2 (and figures S1 and S2) - if it doesn't make the figures too messy and difficult to understand.

Thanks for pointing out the mention in lines 133-134. We prefer not to add land carbon uptake to Supplementary Figure 1 because it overlaps too much with the upper purple region (decrease SO gas exchange rate), which makes the figure too messy. The gas exchange results are more important for our interpretations, while land carbon uptake is only a brief mention (and basically ruled out in the main text). Instead, we added the text "It would appear as the reflection of the green fast land carbon release shading and overlap significantly with the purple region representing decreased Southern Ocean gas exchange rate." to lines 83-85.

2. Figure S2 seems to be missing a label that probably corresponds to a decrease in wind speed in the Bern 3D model.

We verified that all labels are present in Supplementary Figure 2. The label "SO Upwelling (windspeed +20-140%)" refers to 7 squares that plot in a line that cuts through the origin. The confusion probably comes from a mistake in the label. There are two data points that extend to the left and upwards from the origin, which require a reduction in windspeed, but the label "+20-140%" only indicates increase in windspeed. After reviewing Tschumi 2011, we realize we made a mistake in the label. It should read "60-180%," as the 7 points represent windspeed at 60%, 80%, 100%, 120%, 140%, 160%, and 180% relative to a control (the origin). We updated the figure label as well as the description in Supplementary Table 1.

3. I realize that the shaded regions in the cross plots are taken from Bauska et al. (2016), but it's unclear to me how the ranges were calculated (I apologize if I missed this!). Furthermore, not all of the models used in this study fall into these shaded regions (figure S2). Could you expound upon this somewhere?

The shaded regions, taken from Bauska et al. 2016, bound the max/min of the inter- and intra-model variability explored in Bauska 2016 for the last deglaciation. We used these perturbations as a guide for how $\delta^{13}\text{C-CO}_2$ and CO_2 might have changed due to certain processes that would have been active during the period of our study, but built on them with further experimentation. First, we extrapolated the shading backwards (because many processes would have been reversed for MIS 4), ran tests to see if the model was in fact linear when forced in the opposite direction, and added sea ice experiments (Supplementary Figure 1). Then we plotted results from other models (UVic, BICYCLE, Bern3D, LOVECLIM) to see how well they compare. The compilation is meant to be a semi-quantitative guide for how carbon cycle perturbations affect $\delta^{13}\text{C-CO}_2$ and CO_2 concentration, and it is meant to be up-front that not all the models agree. We added text to the beginning of the Figure S2 caption, which now reads, "We took the colored regions from cross-plots in the study of Bauska

et al. 2016³, which bound the inter- and intra-model variability explored by Bauska 2016 for the last deglaciation, and extrapolated the regions backwards to represent perturbations that were likely active during the last glaciation (e.g., cooling SST versus warming).”

Correct, not all of the model results fall within the shaded regions. This is not surprising given the wide range of model geometry, boundary conditions, and experimental designs. Despite some outliers, we think the agreement is generally good between models and supports our model framework. We changed the first two sentences of the final paragraph in this supplementary section (lines 115-117) to read, “While generally the various model results agree in terms of the expected change in $\delta^{13}\text{C-CO}_2$ and CO_2 , there are some results that fall outside of the expected region. This is not surprising given the differences in model geometry, boundary conditions, and experimental design.” In this paragraph we were explicit about the limits of the analysis and feel that further discussion about individual results is outside the scope of the paper.

4. Please double-check that you have appropriately cited all external datasets that you're plotting in figures 1, 3, and 4. I believe there's a reference missing to Eggleston et al. (2015) in figure 1A, for example.

We checked that all datasets are cited correctly in the figure captions for figures 1, 3, and 4. You are correct that we had missed Eggleston 2016, as well as Schmitt 2012!

Reviewer #3 (Remarks to the Author):

Rereview of Menking et al. Multiple carbon cycle mechanisms associated with the glaciation of Marine Isotope Stage 4 (Submission NCOMMS-21-51287A)

I have read the revised submission of Menking et al., and their response to my previous comments and those of the other two reviewers. In my view the authors have responded efficiently and satisfactorily to all points of criticism. I congratulate the authors for their effective revision of the study and for the study itself that will be an important contribution once published and spark new research on the mechanisms driving (the transition in and out of) MIS4 by both, the proxy and modelling community.

Thank you!